

# Transfer matrix spectrum for cyclic representations of the 6-vertex reflection algebra I

**Jean Michel Maillet, Giuliano Niccoli and Baptiste Pezelier**

University of Lyon, ENS de Lyon, University Claude Bernard Lyon 1, CNRS, Laboratoire de Physique, UMR 5672, F-69342 Lyon, France

maillet@ens-lyon.fr, giuliano.niccoli@ens-lyon.fr, baptiste.pezelier@ens-lyon.fr

## Abstract

We study the transfer matrix spectral problem for the cyclic representations of the trigonometric 6-vertex reflection algebra associated to the Bazanov-Stroganov Lax operator. The results apply as well to the spectral analysis of the lattice sine-Gordon model with integrable open boundary conditions. This spectral analysis is developed by implementing the method of separation of variables (SoV). The transfer matrix spectrum (both eigenvalues and eigenstates) is completely characterized in terms of the set of solutions to a discrete system of polynomial equations in a given class of functions. Moreover, we prove an equivalent characterization as the set of solutions to a Baxter's like T-Q functional equation and rewrite the transfer matrix eigenstates in an algebraic Bethe ansatz form. In order to explain our method in a simple case, the present paper is restricted to representations containing one constraint on the boundary parameters and on the parameters of the Bazanov-Stroganov Lax operator. In a next article, some more technical tools (like Baxter's gauge transformations) will be introduced to extend our approach to general integrable boundary conditions.

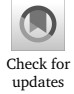

# 1 Introduction

The study of quantum models with integrable open boundary conditions has attracted a large research interest, e.g. see [1–52] and references therein. These models are of physical interest as they can describe both equilibrium and out of equilibrium physics; e.g. some interesting applications concern the description of classical stochastic relaxation processes, like ASEP [53, 54], [48–51], and quantum transport properties in spin systems [55, 56].

In this paper we start the analysis of the class of open integrable quantum models associated to cyclic representations [57–61] of the 6-vertex reflection algebra. The literature of these models is so far rather sparse with the exception of some rather special representations and boundary conditions, like the open XXZ chains at the roots of unity, that can be traced back to these representations under some special constraints, and for which some results are known in the framework of algebraic Bethe ansatz (ABA) [52, 62, 63], fusion of transfer matrices and truncation identities [6, 7, 64–67].

In order to study the general representations and boundary conditions we have to go beyond traditional methods [62–83] which do not apply for these general settings. This is done by developing the Sklyanin's SoV method [84–87] for this class of models, a method that has the advantage to lead (mainly by construction) to the complete characterization of the spectrum (eigenvalues and eigenvectors) and has proven to be applicable for a large variety of integrable quantum models [28–36, 88–102], where traditional methods fail. Moreover, the SoV approach has the advantage to allow also for the study of the dynamics of the models as it leads to universal determinant formulae for matrix elements of local operators on transfer matrix eigenstates as shown for different classes of models, first in [97], and then in many other cases in [33, 88, 89, 99, 100, 103]. Moreover, the analysis developed in [104, 105] makes it possible to compute homogeneous and then thermodynamic limit of these matrix elements opening the way to the computation of the corresponding correlation functions.

Let us recall that in [3], Sklyanin has shown how to construct classes of quantum models with integrable boundaries in the framework of the so-called Quantum Inverse Scattering Method [62–79], constructing in particular associated families of commuting transfer matrices (conserved charges of the model). In fact, Sklyanin's construction allows to use solutions of the Yang-Baxter equation to generate new solutions of the reflection equation [4] once a scalar solution of this last equation is known. Then as a consequence of the reflection equation these new solutions generate commuting transfer matrices [3].

Sklyanin has used the 6-vertex case and the associated XXZ spin 1/2 quantum chains [80, 106–108] to develop an explicit example of this construction. However as pointed out in [3] similar constructions applies also to other 6-vertex cases like the non-linear Schrödinger and the Toda chains as well as for models associated to the 8-vertex case like the XYZ spin 1/2 quantum chains. Further integrable quantum models with open boundary conditions have been presented following the Sklyanin's construction. Interesting examples are the higher spin open quantum chains [17, 18], the higher rank open quantum spin chains [40–42] and the Hubbard model [109–112] with integrable open boundaries [43–48]. In fact, this mainly results in the possibility to associate to any closed integrable quantum model (characterized by a solution of the Yang-Baxter equation) new open integrable quantum models (characterized by the associated Sklyanin's solutions of the reflection equation).

Here, we characterize the spectral problem and we start the analysis of the dynamical problem (by determining scalar products of separate states) for the class of models in the Sklyanin's construction associated to general scalar solutions of the 6-vertex reflection equation [11, 13] and the general Bazhanov-Stroganov cyclic solution of the 6-vertex Yang-Baxter algebra [113].

It may be instructive to recall some main literature on open integrable quantum chains, even for different representations with regards to those studied here. Indeed, this allows us to point out the difficulties that arise in their analysis and that we have also encountered for the models studied in this paper and to give the motivations for the approach that we have followed to overcome them.

The spectrum of the open XXZ spin 1/2 quantum chain with parallel z-oriented magnetic fields on the boundaries has been characterized in [3], in the framework of the algebraic Bethe ansatz. While its dynamics has been studied by the exact computation of correlation functions first in [14, 15] and then in [16], there generalizing in the ABA framework the method established in [114, 115] for the periodic chains. These open quantum spin chains with z-oriented boundary magnetic fields correspond in the Sklyanin's construction to the diagonal scalar solution of the reflection equation. However, the most general scalar solution of the 6-vertex/8-vertex reflection equation is non-diagonal [11, 13] producing unparalleled and not z-oriented boundaries magnetic fields. Under this general setting the analysis of the spectrum and dynamics has shown to be much more involved. The ABA method cannot be directly applied to these open chains with general boundary. In fact, it was first understood in [37], for the XYZ spin 1/2 open chain, that the use of the Baxter's gauge transformations [116] allows to generalize the ABA method limitedly to non-diagonal boundary matrices which satisfy one special constraint. After that, the same approach, based now on the trigonometric version of the Baxter's gauge transformations, was used in [38, 39] to describe the XXZ spectrum by ABA under similar constraints. Let us comment that for the XXZ spectrum the same constraint was derived independently by a pure functional method based on the use of the fusion of transfer matrices and truncations identities for the roots of unit case in [22]- [24]. This has given access to the study of the spectrum leaving however the study of the dynamics for these open models still unsolved.

Results on the spectrum for the most general unconstrained boundary conditions have been achieved only more recently and they have required the introduction of methods different from the ABA. Pure eigenvalue analysis has been implemented in [25] by a functional method leading to nested Bethe ansatz type equations similar to those previously introduced in [26]. Moreover, an ansatz for polynomial T-Q functional equations with an inhomogeneous term has been recently argued in [27]. Eigenstate construction has been first considered under these general boundary in [21] by the q-Onsager algebra formalism. A different approach, based on the generalization of the Sklyanin's separation of variables (SoV) method to the reflection algebra framework, has then lead to the complete eigenvalues and eigenstates characterization [29, 30, 33–36], proving its equivalence to an inhomogeneous TQ functional equation [36],

and also giving access to first computations of matrix elements of local operators [33] in the eigenstates basis.

The aim of the paper is to generalize this type of results for the 6-vertex cyclic representations. Here we solve this problem in the case of one triangular and one general boundary matrix, so that our current results define also the setup for the solution of the most general boundary case as well as the paper [33] has introduced the tools to solve the case with the most general boundary in [35].

The paper is organized as it follows. In section 2, we recall the cyclic representations of the 6-vertex Yang-Baxter algebra associated to the Bazhanov-Stroganov Lax-operator. In section 3, we define the associated representations of the cyclic 6-vertex reflection algebra. In section 4, we prove the diagonalizability of the generator $\mathscr{B}_-(\lambda)$ of the reflection algebra generated by $\mathscr{U}_-(\lambda)$ for the most general $K_-(\lambda)$ boundary matrix while we impose one constraint on the parameters of the Bazhanov-Stroganov Lax-operator for any quantum site to make easier the explicit construction of the $\mathscr{B}_-$-eigenstates basis. Moreover, we compute the scalar product for the so-called separate states in the $\mathscr{B}_-$-eigenstates basis. In section 5, we show that the $\mathscr{B}_-$-eigenstates basis is the SoV-basis for the transfer matrix spectral problem associated to the most general $K_-$-boundary matrix and upper triangular $K_+$-boundary matrix and we solve in this SoV basis this spectral problem. In section 6, we show that the SoV characterization of the transfer matrix spectrum is equivalent to inhomogeneous Baxter's like TQ-functional equation with polynomial Q-functions. We present four appendices, in the first one we extend the proof of section 4 for the diagonalizability of the operator $\mathscr{B}_-(\lambda)$ to the case of general values of the boundary and bulk parameters. The remaining three appendices deal with the reduction of our representations to those associated to the chiral-Potts, the sine-Gordon and the XXZ spin s-chains at the 2s+1 roots of unit.

## 2 Cyclic representations of the 6-vertex Yang-Baxter algebra

In this section we recall the basics of the cyclic representations of the 6-vertex Yang-Baxter algebra associated to the Bazhanov-Stroganov Lax-operator. We consider the representations defined by the tensor product of N local representations of the 6-vertex Yang-Baxter algebra on the local Hilbert spaces $\mathscr{R}_n$. Each local representation is defined as the representation of a local Weyl algebra

$$u_n v_m = q^{\delta_{n,m}} v_m u_n \quad \forall n, m \in \{1, ..., N\}, \tag{2.1}$$

associated to a root of unit $q$, where $u_n$ and $v_n$ are the Weyl algebra generators on the Hilbert spaces $\mathscr{R}_n$. Here, we assume that $u_n$ and $v_n$ are unitary operators and that it holds:

$$u_n^p = v_n^p = 1 \quad \text{for} \quad q = e^{-i\pi\beta^2}, \tag{2.2}$$

with $\beta^2 = p'/p$, $p'$ even and $p = 2l + 1$ odd. This type of representation can be defined on a $p$-dimensional linear space $\mathscr{R}_n$, imposing that the $v_n$ spectrum coincides with the $p$-roots of the unit:

$$v_n|k, n\rangle = q^k|k, n\rangle \quad \forall (n, k) \in \{1, ..., N\} \times \{-l, ..., l\}. \tag{2.3}$$

On $\mathscr{R}_n$ is defined a $p$-dimensional representation of the Weyl algebra by setting:

$$u_n|k, n\rangle = |k + 1, n\rangle \quad \forall k \in \{-l, ..., l\}, \tag{2.4}$$

with the cyclicity condition:

$$|k + p, n\rangle = |k, n\rangle. \tag{2.5}$$

$\mathcal{R}_n$ is also called the right local quantum space at the site $n$ of the chain. Let $\mathcal{L}_n$ be the dual space of $\mathcal{R}_n$ then we can define the following scalar products:

$$\langle k, n | k', n \rangle = ((\langle k, n |)^\dagger, | k', n \rangle) \equiv \delta_{k,k'}, \tag{2.6}$$

for any $k, k' \in \{-l, ..., l\}$.

The local generators of the cyclic 6-vertex Yang-Baxter algebra can be now defined as the elements of the following Bazhanov-Stroganov Lax operator:

$$L_{a,n}(\lambda) \equiv \begin{pmatrix} \lambda \alpha_n \nu_n - \beta_n \lambda^{-1} \nu_n^{-1} & u_n \left( q^{-1/2} a_n \nu_n + q^{1/2} b_n \nu_n^{-1} \right) \\ u_n^{-1} \left( q^{1/2} c_n \nu_n + q^{-1/2} d_n \nu_n^{-1} \right) & \gamma_n \nu_n / \lambda - \delta_n \lambda / \nu_n \end{pmatrix}_a \in \mathrm{End}(\mathbb{C}^2 \otimes \mathcal{R}_n), \tag{2.7}$$

where $a$ denote the so-called auxiliary space $V_a \simeq \mathbb{C}^2$. Indeed, under the condition

$$\gamma_n = a_n c_n / \alpha_n, \qquad \delta_n = b_n d_n / \beta_n, \tag{2.8}$$

$L_{a,n}(\lambda)$ is a solution of the 6-vertex Yang-Baxter equation:

$$R_{12}(\lambda/\mu) L_{1,n}(\lambda) L_{2,n}(\mu) = L_{2,n}(\mu) L_{1,n}(\lambda) R_{12}(\lambda/\mu), \tag{2.9}$$

with regards to the standard 6-vertex $R$-matrix:

$$R_{ab}(\lambda) = \begin{pmatrix} q\lambda - q^{-1}\lambda^{-1} & & & \\ & \lambda - \lambda^{-1} & q - q^{-1} & \\ & q - q^{-1} & \lambda - \lambda^{-1} & \\ & & & q\lambda - q^{-1}\lambda^{-1} \end{pmatrix}, \tag{2.10}$$

where $a$ and $b$ denote two bidimensional spaces $V_a, V_b \equiv \mathbb{C}^2$ and $R_{ab}(\lambda)$ is an endomorphism on their tensor product, i.e. $R_{ab}(\lambda) \in \mathrm{End}(\mathbb{C}^2 \otimes \mathbb{C}^2)$. Then, the following monodromy matrix:

$$M_a(\lambda) = \begin{pmatrix} A(\lambda) & B(\lambda) \\ C(\lambda) & D(\lambda) \end{pmatrix}_a \equiv L_{a,\mathsf{N}}(\lambda q^{-1/2}) \cdots L_{a,1}(\lambda q^{-1/2}) \in \mathrm{End}(\mathbb{C}^2 \otimes \mathcal{H}), \tag{2.11}$$

is also a solution of the Yang-Baxter equation:

$$R_{12}(\lambda/\mu) M_1(\lambda) M_2(\mu) = M_2(\mu) M_1(\lambda) R_{12}(\lambda/\mu), \tag{2.12}$$

and its elements define a representation of the Yang-Baxter algebra on the tensor product of the local representation spaces, i.e. $\mathcal{H} = \otimes_{n=1}^{\mathsf{N}} \mathcal{R}_n$. Note that one can also consider cyclic representations of the 6-vertex Yang-Baxter algebra associated to $q$, an even root of unit, these have been studied in [117].

## 2.1 Bulk transfer matrix and quantum determinant

The Yang-Baxter equation implies that the bulk transfer matrix $\tau_2(\lambda) \equiv \mathrm{tr}_a M_a(\lambda)$ defines a one parameter family of commuting operators. Note that we have:

$$[\tau_2(\lambda), \Theta] = 0, \quad \text{where} \quad \Theta \equiv \prod_{n=1}^{\mathsf{N}} \nu_n. \tag{2.13}$$

In [113, 118] it was related to the analysis of the chP-model [119–126] and characterized by SoV in [92–97]. The Yang-Baxter equation also implies that the so-called quantum determinant is a central element and it has the following factorized form:

$$\det_q M_a(\lambda) \equiv A(\lambda q^{1/2}) D(\lambda q^{-1/2}) - B(\lambda q^{1/2}) C(\lambda q^{-1/2}) \tag{2.14}$$

$$= D(\lambda q^{1/2}) A(\lambda q^{-1/2}) - C(\lambda q^{1/2}) B(\lambda q^{-1/2}) \tag{2.15}$$

$$= \prod_{n=1}^{\mathsf{N}} \det_q L_{a,n}(\lambda), \tag{2.16}$$

where the local quantum determinants read:

$$\det_q L_{a,n}(\lambda) \equiv \big(L_{a,n}(\lambda)\big)_{11}\big(L_{a,n}(\lambda q^{-1})\big)_{22} - \big(L_{a,n}(\lambda)\big)_{12}\big(L_{a,n}(\lambda q^{-1})\big)_{21} \tag{2.17}$$
$$= \big(L_{a,n}(\lambda)\big)_{22}\big(L_{a,n}(\lambda q^{-1})\big)_{11} - \big(L_{a,n}(\lambda)\big)_{21}\big(L_{a,n}(\lambda q^{-1})\big)_{12}. \tag{2.18}$$

They admit the following explicit form:

$$\det_q M_a(\lambda) = \prod_{n=1}^{N} k_n \big(\frac{\lambda}{\mu_{n,+}} - \frac{\mu_{n,+}}{\lambda}\big)\big(\frac{\lambda}{\mu_{n,-}} - \frac{\mu_{n,-}}{\lambda}\big) \tag{2.19}$$

$$= (-q)^N \prod_{n=1}^{N} \frac{\beta_n a_n c_n}{\alpha_n}\big(\frac{1}{\lambda} + q^{-1}\frac{b_n \alpha_n}{a_n \beta_n}\lambda\big)\big(\frac{1}{\lambda} + q^{-1}\frac{d_n \alpha_n}{c_n \beta_n}\lambda\big) \tag{2.20}$$

$$= a(\lambda)d(\lambda/q), \tag{2.21}$$

where:

$$k_n \equiv (a_n b_n c_n d_n)^{1/2}, \quad \mu_{n,h} \equiv \begin{cases} iq^{1/2}(a_n \beta_n/\alpha_n b_n)^{1/2} & h = +, \\ iq^{1/2}(c_n \beta_n/\alpha_n d_n)^{1/2} & h = -. \end{cases} \tag{2.22}$$

$$a(\lambda) \equiv a_0 \prod_{n=1}^{N}\big(\frac{\beta_n}{\lambda} + q^{-1}\frac{b_n \alpha_n}{a_n}\lambda\big), \quad d(\lambda) \equiv \frac{(-1)^N}{a_0}\prod_{n=1}^{N}\frac{a_n c_n}{\alpha_n}\big(\frac{1}{\lambda} + q\frac{d_n \alpha_n}{c_n \beta_n}\lambda\big), \tag{2.23}$$

and $a_0$ is a free non zero parameter.

# 3 Cyclic representations of the 6-vertex reflection algebra

In this section we define the most general cyclic representations of the 6-vertex reflection algebra associated to the Bazhanov-Stroganov Lax-operator. This is done by following the general procedure introduced by Sklyanin [3] which us allows to associate to any solution $M_a(\lambda) \in \text{End}(\mathbb{C}^2 \otimes \mathcal{H})$ of the 6-vertex Yang-Baxter equation a solution $\mathcal{U}_a(\lambda) \in \text{End}(\mathbb{C}^2 \otimes \mathcal{H})$ of the 6-vertex reflection equation:

$$R_{12}(\lambda/\mu)\,\mathcal{U}_1(\lambda)R_{12}(\lambda\mu/q)\,\mathcal{U}_2(\mu) = \mathcal{U}_2(\mu)R_{12}(\lambda\mu/q)\,\mathcal{U}_1(\lambda)R_{12}(\lambda/\mu). \tag{3.1}$$

Here, we have defined:

$$\mathcal{U}_a(\lambda) = M_a(\lambda)K_a(\lambda)\hat{M}_a(\lambda), \tag{3.2}$$

where $K_a(\lambda; \zeta, \kappa, \tau)$ is the most general scalar (boundary matrix) solution of the 6-vertex reflection equation [11, 12]:

$$K_a(\lambda; \zeta, \kappa, \tau) = \frac{1}{\zeta - \frac{1}{\zeta}}\begin{pmatrix} \frac{\lambda\zeta}{q^{1/2}} - \frac{q^{1/2}}{\lambda\zeta} & \kappa e^{\tau}\big(\frac{\lambda^2}{q} - \frac{q}{\lambda^2}\big) \\ \kappa e^{-\tau}\big(\frac{\lambda^2}{q} - \frac{q}{\lambda^2}\big) & \frac{q^{1/2}\zeta}{\lambda} - \frac{\lambda}{\zeta q^{1/2}} \end{pmatrix}_a, \tag{3.3}$$

and we have defined:

$$\hat{M}_a(\lambda) = (-1)^N \sigma_a^y M_a^{t_a}(1/\lambda)\sigma_a^y. \tag{3.4}$$

Using this correspondence, the most general cyclic representations of the 6-vertex Yang-Baxter algebra, associated to the bulk monodromy matrix (2.11), define the most general cyclic representations of the 6-vertex reflection algebra, corresponding to the boundary monodromy matrices:

$$\mathcal{U}_{a,-}(\lambda) = M_a(\lambda)K_{a,-}(\lambda)\hat{M}_a(\lambda) = \begin{pmatrix} \mathcal{A}_-(\lambda) & \mathcal{B}_-(\lambda) \\ \mathcal{C}_-(\lambda) & \mathcal{D}_-(\lambda) \end{pmatrix}_a, \tag{3.5}$$

$$\mathcal{U}_{a,+}^{t_a}(\lambda) = M_a^{t_a}(\lambda)K_{a,+}^{t_a}(\lambda)\hat{M}_a^{t_a}(\lambda) = \begin{pmatrix} \mathcal{A}_+(\lambda) & \mathcal{C}_+(\lambda) \\ \mathcal{B}_+(\lambda) & \mathcal{D}_+(\lambda) \end{pmatrix}_a, \tag{3.6}$$

with $\mathscr{U}_{a,-}(\lambda)$ and $\mathscr{V}_{a,+}(\lambda) \equiv \mathscr{U}_{a,+}^{t_a}(-\lambda)$ both solution of the reflection equation (3.1), where:

$$K_{a,\pm}(\lambda) = K_a(\lambda q^{(1\pm1)/2}; \zeta_\pm, \kappa_\pm, \tau_\pm) = \begin{pmatrix} a_\pm(\lambda) & b_\pm(\lambda) \\ c_\pm(\lambda) & d_\pm(\lambda) \end{pmatrix}_a, \qquad (3.7)$$

and $\zeta_\pm, \delta_\pm, \tau_\pm$ are arbitrary complex parameters.

## 3.1 Boundary transfer matrix and quantum determinant

These boundary monodromy matrices define a one parameter family of commuting transfer matrices:

$$\mathscr{T}(\lambda) \equiv \mathrm{tr}_a\{K_{a,+}(\lambda)M_a(\lambda)K_{a,-}(\lambda)\hat{M}_a(\lambda)\} \qquad (3.8)$$

$$= \mathrm{tr}_a\{K_{a,-}(\lambda)\mathscr{U}_{a,+}(\lambda)\} = \mathrm{tr}_a\{K_{a,+}(\lambda)\mathscr{U}_{a,-}(\lambda)\} \qquad (3.9)$$

$$= a_+(\lambda)\mathscr{A}_-(\lambda) + d_+(\lambda)\mathscr{D}_-(\lambda) + b_+(\lambda)\mathscr{C}_-(\lambda) + c_+(\lambda)\mathscr{B}_-(\lambda). \qquad (3.10)$$

This statement follows by using the reflection equation as Sklyanin has proven in [3]. The characterization of the spectrum (eigenvalues and eigenstates) of this class of transfer matrices is the main subject of this paper. In particular, we will restrict our attention to the special boundary condition $b_+(\lambda) = 0$, which can be analyzed by implementing the SoV approach once is proven the diagonalizability of the $\mathscr{B}_-(\lambda)$ family of commuting operators. In order to introduce this spectral analysis we start pointing out some important properties satisfied by the generators of the reflection algebra $\mathscr{A}_-(\lambda)$, $\mathscr{B}_-(\lambda)$, $\mathscr{C}_-(\lambda)$ and $\mathscr{D}_-(\lambda)$.

Let us start with the following re-parametrization of the boundary parameters [22]:

$$(\alpha_- - 1/\alpha_-)(\beta_- + 1/\beta_-) \equiv \frac{\zeta_- - 1/\zeta_-}{\kappa_-}, \quad (\alpha_- + 1/\alpha_-)(\beta_- - 1/\beta_-) \equiv \frac{\zeta_- + 1/\zeta_-}{\kappa_-}. \qquad (3.11)$$

Then we define the following functions:

$$\mathsf{A}_-(\lambda) \equiv g_-(\lambda)a(\lambda q^{-1/2})d(1/(q^{1/2}\lambda)), \qquad (3.12)$$

where:

$$g_-(\lambda) \equiv \frac{(\lambda\alpha_-/q^{1/2} - q^{1/2}/(\lambda\alpha_-))(\lambda\beta_-/q^{1/2} + q^{1/2}/(\lambda\beta_-))}{(\alpha_- - 1/\alpha_-)(\beta_- + 1/\beta_-)}. \qquad (3.13)$$

**Proposition 3.1.** *The following boundary quantum determinant:*

$$det_q \mathscr{U}_{a,-}(\lambda) \equiv ((\lambda/q)^2 - (q/\lambda)^2)[\mathscr{A}_-(\lambda q^{1/2}).\mathscr{A}_-(q^{1/2}/\lambda) + \mathscr{B}_-(\lambda q^{1/2})\mathscr{C}_-(q^{1/2}/\lambda)] \quad (3.14)$$

$$= ((\lambda/q)^2 - (q/\lambda)^2)[\mathscr{D}_-(\lambda q^{1/2})\mathscr{D}_-(q^{1/2}/\lambda) + \mathscr{C}_-(\lambda q^{1/2})\mathscr{B}_-(q^{1/2}/\lambda)], \quad (3.15)$$

*is a central element in the reflection algebra, i.e.*

$$[det_q \mathscr{U}_{a,-}(\lambda), \mathscr{U}_{a,-}(\mu)] = 0, \qquad (3.16)$$

*and its explicit expression reads:*

$$det_q \mathscr{U}_{a,-}(\lambda) = (\lambda^2/q^2 - q^2/\lambda^2)\mathsf{A}_-(\lambda q^{1/2})\mathsf{A}_-(q^{1/2}/\lambda). \qquad (3.17)$$

*Moreover, the generators of the reflection algebra satisfy the following properties:*

$$\mathscr{D}_-(\lambda) = \frac{(\lambda^2/q - q/\lambda^2)}{(\lambda^2 - 1/\lambda^2)}\mathscr{A}_-(\lambda^{-1}) + \frac{(q - 1/q)}{(\lambda^2 - 1/\lambda^2)}\mathscr{A}_-(\lambda), \qquad (3.18)$$

*and*

$$\mathscr{B}_-(\lambda^{-1}) = -\frac{(\lambda^2 q - 1/(q\lambda^2))}{(\lambda^2/q - q/\lambda^2)}\mathscr{B}_-(\lambda), \quad \mathscr{C}_-(\lambda^{-1}) = -\frac{(\lambda^2 q - 1/(q\lambda^2))}{(\lambda^2/q - q/\lambda^2)}\mathscr{C}_-(\lambda). \qquad (3.19)$$

We omit the proof of this proposition as it can be derived repeating the main steps of the original Sklyanin's paper, where similar statements were proven for the case of spin $1/2$ representations of the 6-vertex reflection algebra. Let us introduce now the following notation:

$$\mathscr{T}_{\diagdown}(\lambda) \equiv \mathsf{a}_+(\lambda)\,\mathscr{A}_-(\lambda) + \mathsf{d}_+(\lambda)\,\mathscr{D}_-(\lambda), \tag{3.20}$$

for the diagonal part of the transfer matrix, i.e. the one associated to the diagonal elements of the matrix $\mathscr{U}_{a,-}(\lambda)$, and the coefficients:

$$\mathsf{a}_+(\lambda) \equiv \frac{(\lambda^2 q - 1/(q\lambda^2))(\lambda\zeta_+/q^{1/2} - q^{1/2}/(\lambda\zeta_+))}{(\lambda^2 - 1/\lambda^2)(\zeta_+ - 1/\zeta_+)}, \tag{3.21}$$

$$\mathsf{d}_+(\lambda) \equiv \frac{(\lambda^2 q - 1/(q\lambda^2))(\zeta_+ q^{1/2}/\lambda - \lambda/(q^{1/2}\zeta_+))}{(\lambda^2 - 1/\lambda^2)(\zeta_+ - 1/\zeta_+)}. \tag{3.22}$$

then we can prove the following:

**Corollary 3.1.** *The most general transfer matrix admits the following symmetries:*

$$\mathscr{T}(\lambda) = \mathscr{T}(1/\lambda), \quad \mathscr{T}(-\lambda) = \mathscr{T}(\lambda), \tag{3.23}$$

*and the diagonal part $\mathscr{T}_{\diagdown}(\lambda)$ has the following explicitly symmetric forms:*

$$\mathscr{T}_{\diagdown}(\lambda) \equiv \mathsf{a}_+(\lambda)\mathscr{A}_-(\lambda) + \mathsf{a}_+(1/\lambda)\mathscr{A}_-(1/\lambda) \tag{3.24}$$

$$= \mathsf{d}_+(\lambda)\mathscr{D}_-(\lambda) + \mathsf{d}_+(1/\lambda)\mathscr{D}_-(1/\lambda). \tag{3.25}$$

# 4 SoV representation of cyclic 6-vertex reflection algebra

In this section we construct the left and right basis which diagonalize the one-parameter family of commuting operators $\mathscr{B}_-(\lambda)$ associated to the most general $K_-(\lambda)$ matrix. Here we impose one constraint on the parameters of the representation at any quantum site:

$$b_n^p + a_n^p = 0, \quad \forall n \in \{1, ..., \mathsf{N}\}. \tag{4.1}$$

This is done in order to make completely explicit the construction of this basis; however, the proof of the diagonalizability of $\mathscr{B}_-(\lambda)$ can be done without these constraints and under completly general values of the inner boundary matrix and of the bulk parameters and it will be presented in appendix.

## 4.1 Pseudo-vacuum states

We implement the above constraints by imposing:

$$b_n = -q^{2j_n - 1}a_n, \tag{4.2}$$

where for any $n \in \{1, ..., \mathsf{N}\}$ we have fixed $j_n \in \{0, ..., p-1\}$, then we have:

$$\langle j_n - 1, n|\left(L_{a,n}\right)_{12} = \underline{0}, \quad \left(L_{a,n}\right)_{12}|j_n, n\rangle = \underline{0}, \tag{4.3}$$

as well as:

$$\langle j_n - 1, n|\left(L_{a,n}(\lambda)\right)_{11} = \mathsf{a}_n(\lambda q^{j_n-1})\langle j_n - 1, n| \tag{4.4}$$

$$\langle j_n - 1, n|\left(L_{a,n}(\lambda)\right)_{22} = \mathsf{d}_n(\lambda q^{1-j_n})\langle j_n - 1, n|, \tag{4.5}$$

and

$$\left(L_{a,n}(\lambda)\right)_{11}|j_n,n\rangle = |j_n,n\rangle\, a_n(\lambda q^{j_n}), \qquad \left(L_{a,n}(\lambda)\right)_{22}|j_n,n\rangle = |j_n,n\rangle\, d_n(\lambda q^{-j_n}), \tag{4.6}$$

where:

$$a_n(\lambda) = \lambda\alpha_n - \beta_n/\lambda, \quad d_n(\lambda) = \gamma_n/\lambda - \lambda\delta_n, \tag{4.7}$$

which is of course compatible with the local quantum determinant at site $n$:

$$\langle j_n-1,n|\det_q L_{a,n}(\lambda) = \langle j_n-1,n|\left[\left(L_{a,n}(\lambda)\right)_{11}\left(L_{a,n}(\lambda/q)\right)_{22} - \left(L_{a,n}\right)_{12}\left(L_{a,n}\right)_{21}\right] \tag{4.8}$$

$$= a_n(\lambda q^{j_n-1})d_n(\lambda q^{-j_n})\langle j_n-1,n| \tag{4.9}$$

$$\det_q L_{a,n}(\lambda)|j_n,n\rangle = \left[\left(L_{a,n}(\lambda)\right)_{22}\left(L_{a,n}(\lambda/q)\right)_{11} - \left(L_{a,n}\right)_{21}\left(L_{a,n}\right)_{12}\right]|j_n,n\rangle \tag{4.10}$$

$$= |j_n,n\rangle\, a_n(\lambda q^{j_n-1})d_n(\lambda q^{-j_n}), \tag{4.11}$$

being:

$$a_n(\lambda q^{j_n-1})d_n(\lambda q^{-j_n}) = -q\frac{\beta_n a_n c_n}{\alpha_n}\left(\frac{1}{\lambda} - q^{2(j_n-1)}\frac{\alpha_n}{\beta_n}\lambda\right)\left(\frac{1}{\lambda} + q^{-1}\frac{d_n\alpha_n}{c_n\beta_n}\lambda\right). \tag{4.12}$$

Then we can define the following left and right "reference states":

$$\langle\Omega| = \otimes_{n=1}^{N}\langle j_n-1,n|, \quad |\bar{\Omega}\rangle = \otimes_{n=1}^{N}|j_n,n\rangle. \tag{4.13}$$

The following properties are satisfied:

$$\langle\Omega|A(\lambda q^{1/2}) = a(\lambda)\langle\Omega|, \quad \langle\Omega|D(\lambda q^{1/2}) = d(\lambda)\langle\Omega|, \quad \langle\Omega|B(\lambda) = 0, \quad \langle\Omega|C(\lambda) \neq 0, \tag{4.14}$$

$$A(\lambda q^{1/2})|\bar{\Omega}\rangle = |\bar{\Omega}\rangle a(\lambda q), \quad D(\lambda q^{1/2})|\bar{\Omega}\rangle = |\bar{\Omega}\rangle d(\lambda/q), \quad B(\lambda)|\bar{\Omega}\rangle = 0, \quad C(\lambda)|\bar{\Omega}\rangle \neq 0, \tag{4.15}$$

where it is simple to verify that as it should:

$$a(\lambda) = \prod_{n=1}^{N} a_n(\lambda q^{j_n-1}), \quad d(\lambda) = \prod_{n=1}^{N} d_n(\lambda q^{1-j_n}), \tag{4.16}$$

once we have fixed the free parameter:

$$a_0 = (-q)^N \prod_{n=1}^{N} q^{-j_n}. $$

Of course, the coefficients $a(\lambda)$ and $d(\lambda)$ as well as the reference states depend on the choice of the N-tuple $\{j_1,...,j_N\}$ but for simplicity we do not write it explicitly.

## 4.2 Representation of the reflection algebra in $\mathscr{B}_-(\lambda)$-eigenstates basis

The left and right SoV-representations of the cyclic 6-vertex reflection algebra are now defined by constructing the left and right $\mathscr{B}_-(\lambda)$-eigenstates basis and by determining in this new basis the representation of the other generators of the algebra. In order to present our results we need to introduce some notations. We define the following functions parametrized by the $N$-tuples $\boldsymbol{h} \equiv (h_1,...,h_N) \in \{0,...,p-1\}^N$:

$$\mathrm{B}_{\boldsymbol{h}}(\lambda) \equiv \kappa_- e^{\tau_-}\frac{(\lambda^2/q - q/\lambda^2)}{(\zeta_- - 1/\zeta_-)}a_{\boldsymbol{h}}(\lambda)a_{\boldsymbol{h}}(1/\lambda), \tag{4.17}$$

with

$$a_{\boldsymbol{h}}(\lambda) \equiv (-1)^N \prod_{n=1}^{N}(\alpha_n\beta_n)^{1/2}\left(\frac{\lambda}{\xi_n^{(h_n)}} - \frac{\xi_n^{(h_n)}}{\lambda}\right), \tag{4.18}$$

where:

$$\xi_n^{(h)} = \mu_{n,+} q^{h+1/2}, \ \xi_{n+\mathsf{N}}^{(h)} \equiv \xi_n^{(h)} \ \forall n \in \{1, ..., \mathsf{N}\}, \ a_0(\lambda) = a(\lambda/q^{1/2}). \tag{4.19}$$

Moreover, next, we will need also the following notations:

$$\Lambda = (\lambda^2 + 1/\lambda^2), \quad X_b^{(h_b)} = (\zeta_b^{(h_b)})^2 + 1/(\zeta_b^{(h_b)})^2, \ X = q + 1/q \tag{4.20}$$

$$\zeta_n^{(h)} = \left(\xi_n^{(h)}\right)^{\varphi_n} \ \text{for} \ h \in \{0, ..., p-1\} \ \text{and} \ \forall n \in \{1, ..., 2\mathsf{N}\}, \tag{4.21}$$

$$\varphi_a = 1 - 2\theta(a - \mathsf{N}) \ \text{with} \ \theta(x) = \{0 \text{ for } x \leq 0, \ 1 \text{ for } x > 0\}. \tag{4.22}$$

**Theorem 4.1** (Left $\mathscr{B}_-(\lambda)$ SOV-representations). *If $b_-(\lambda) \neq 0$ and it holds:*

$$\mu_{n,+}^p \neq \mu_{m,+}^p \quad \forall n \neq m \in \{1, ..., \mathsf{N}\}, \tag{4.23}$$

*and*

$$\mu_{n,+}^{2p} \neq \pm 1, \ \mu_{n,+}^2 \neq q^{-2h}\alpha_-^{2\epsilon}, \ \mu_{n,+}^2 \neq -q^{-2h}\beta_-^{2\epsilon}, \ \mu_{n,+}^2 \neq q^{-2\epsilon-2h}\mu_{m,-}^{2\epsilon} \tag{4.24}$$

*for any $\epsilon = \pm 1$, $h \in \{1, ..., p-1\}$ and $n, m \in \{1, ..., \mathsf{N}\}$, then the states:*

$$\langle h_1, ..., h_\mathsf{N}| \equiv \frac{1}{\mathsf{N}} \langle \Omega| \prod_{n=1}^{\mathsf{N}} \prod_{k_n=1}^{h_n} \frac{\mathscr{A}_-(1/\xi_n^{(k_n-1)})}{\mathsf{A}_-(1/\xi_n^{(k_n-1)})}, \tag{4.25}$$

*where $h_n \in \{0, ..., p-1\}$ and $\mathsf{N}$ is a free normalization, define a $\mathscr{B}_-(\lambda)$-eigenstates basis of $\mathscr{H}^*$:*

$$\langle \boldsymbol{h}|\mathscr{B}_-(\lambda) = \mathsf{B}_{\boldsymbol{h}}(\lambda)\langle \boldsymbol{h}|. \tag{4.26}$$

*Here we have denoted $\langle \boldsymbol{h}| \equiv \langle h_1, ..., h_\mathsf{N}|$. The quantum determinant and the following left action on the generic state $\langle \boldsymbol{h}|$:*

$$\langle \boldsymbol{h}|\mathscr{A}_-(\lambda) = \sum_{a=1}^{2\mathsf{N}} \frac{(\lambda^2/q - q/\lambda^2)(\lambda\zeta_a^{(h_a)} - 1/\zeta_a^{(h_a)}\lambda)}{((\zeta_a^{(h_a)})^2/q - q/(\zeta_a^{(h_a)})^2)((\zeta_a^{(h_a)})^2 - 1/(\zeta_a^{(h_a)})^2)} \prod_{\substack{b=1 \\ b \neq a \ mod\mathsf{N}}}^{\mathsf{N}} \frac{\Lambda - X_b^{(h_b)}}{X_a^{(h_a)} - X_b^{(h_b)}} \mathsf{A}_-(\zeta_a^{(h_a)})$$

$$\times \ \langle \boldsymbol{h}|T_a^{-\varphi_a} + (-1)^\mathsf{N} det_q M(1) \frac{(\lambda/q^{1/2} + q^{1/2}/\lambda)}{2} \prod_{b=1}^{\mathsf{N}} \frac{\Lambda - X_b^{(h_b)}}{X - X_b^{(h_b)}} \langle \boldsymbol{h}|$$

$$+ (-1)^\mathsf{N} \frac{(\zeta_- + 1/\zeta_-)}{(\zeta_- - 1/\zeta_-)} det_q M(i) \frac{(\lambda/q^{1/2} - q^{1/2}/\lambda)}{2} \prod_{b=1}^{\mathsf{N}} \frac{\Lambda - X_b^{(h_b)}}{X + X_b^{(h_b)}} \langle \boldsymbol{h}|, \tag{4.27}$$

*where:*

$$\langle h_1, ..., h_a, ..., h_\mathsf{N}|T_a^\pm = \langle h_1, ..., h_a \pm 1, ..., h_\mathsf{N}|, \tag{4.28}$$

*completely determine the representation of the other generators of the reflection algebra in the $\mathscr{B}_-(\lambda)$-eigenstates basis. Indeed, the representation of $\mathscr{D}_-(\lambda)$ follows from the identity (3.18) while $\mathscr{C}_-(\lambda)$ by the quantum determinant.*

*Proof.* Let us write explicitly the decomposition of the reflection algebra generator:

$$\mathscr{B}_-(\lambda) = (-1)^\mathsf{N}[-a_-(\lambda)A(\lambda)B(1/\lambda) + b_-(\lambda)A(\lambda)A(1/\lambda)$$
$$- c_-(\lambda)B(\lambda)B(1/\lambda) + d_-(\lambda)B(\lambda)A(1/\lambda)], \tag{4.29}$$

in terms of the generators of the Yang-Baxter algebra. Then, by using the identities (4.14) it follows that $\langle \Omega|$ is a $\mathscr{B}_-(\lambda)$-eigenstate with non-zero eigenvalue:

$$\langle \Omega| \mathscr{B}_-(\lambda) \equiv \mathsf{B}_0(\lambda) \langle \Omega|. \tag{4.30}$$

Now by using the reflection algebra commutation relations:

$$\mathscr{A}_-(\lambda_2)\mathscr{B}_-(\lambda_1) = \frac{(\lambda_1 q/\lambda_2 - \lambda_2/(\lambda_1 q))(\lambda_1\lambda_2/q - q/(\lambda_1\lambda_2))}{(\lambda_1/\lambda_2 - \lambda_2/\lambda_1)(\lambda_1\lambda_2 - 1/(\lambda_1\lambda_2))}\mathscr{B}_-(\lambda_1).\mathscr{A}_-(\lambda_2)$$
$$+ \frac{(\lambda_1^2/q - q/\lambda_1^2)(q - 1/q)}{(\lambda_2/\lambda_1 - \lambda_1/\lambda_2)(\lambda_1^2 - 1/\lambda_1^2)}\mathscr{B}_-(\lambda_2).\mathscr{A}_-(\lambda_1)$$
$$- \frac{q - 1/q}{(\lambda_1^2 - 1/\lambda_1^2)(\lambda_1\lambda_2 - 1/(\lambda_1\lambda_2))}\mathscr{B}_-(\lambda_2)\tilde{\mathscr{D}}_-(\lambda_1) \qquad (4.31)$$

we can follow step by step the proof given in [99] to prove the validity of (4.26). The action of $\mathscr{A}_-(\zeta_b^{(h_b)})$ for $b \in \{1, ..., 2N\}$ follows from the definition of the states $\langle \mathbf{h}|$, the reflection algebra commutation relations (4.31) and the quantum determinant relations. Let us show now that the conditions (4.23) and (4.24) imply that the set of $p^N$ states $\langle \mathbf{h}|$ is a $\mathscr{B}_-(\lambda)$-eigenstates basis of $\mathscr{H}^*$. As by condition (4.23) each such state is associated to a different eigenvalue of $\mathscr{B}_-(\lambda)$ the only thing that we need to prove to get their linear independence is that each such state is nonzero. We know by construction that the state $\langle \Omega|$ is nonzero so let us assume by induction that the same is true for the state $\langle \mathbf{h}^{(0)}| = \langle h_1^{(0)}, ..., h_N^{(0)}|$ with $h_j^{(0)} \in \{0, ..., p-2\}$ and let us show that $\langle \mathbf{h}_j^{(0)}| = \langle h_1^{(0)}, ..., h_j^{(0)} + 1, ..., h_N^{(0)}|$ is nonzero. We have that:

$$\langle \mathbf{h}_j^{(0)}|\mathscr{A}_-(\xi_j^{(h_j^{(0)}+1)}) = A_-(\xi_j^{(h_j^{(0)}+1)})\langle \mathbf{h}^{(0)}| \neq \underline{0} \qquad j \in \{1, ..., N\} \qquad (4.32)$$

so that $\langle \mathbf{h}_j^{(0)}|$ is nonzero. Using this we can prove that all states $\langle \mathbf{h}^{(1)}| = \langle h_1^{(0)} + x_1, ..., h_N^{(0)} + x_N|$ with $x_j \in \{0, 1\}$ for any $j \in \{1, ..., N\}$ are nonzero, which just prove the validity of the induction. Finally, by using the identities:

$$\mathscr{U}_-(q^{1/2}) = (-1)^N \det_q M(1) \, I_0, \quad \mathscr{U}_-(iq^{1/2}) = i(-1)^{N+1}\frac{\zeta_- + 1/\zeta_-}{\zeta_- - 1/\zeta_-}\det_q M(i) \, \sigma_0^z, \qquad (4.33)$$

and remarking that $\mathscr{A}_-(\lambda)$ has the following functional dependence with regards to $\lambda$:

$$\mathscr{A}_-(\lambda) = \sum_{a=0}^{2N+1} \lambda^{(2a-2N+1)}\mathscr{A}_{-,a}, \qquad (4.34)$$

where $\mathscr{A}_{-,a} \in \text{End}(\mathscr{H})$ are some fixed operators, we get our interpolation formula for its action on $\langle \mathbf{h}|$. $\qquad \square$

Similarly, defining:

$$\kappa_a^{(h)} = k(\zeta_a^{(h)}), \quad \text{for } h \in \{0, ..., p-1\}, \; a \in \{1, ..., 2N\}, \qquad (4.35)$$

and the function:

$$k(\lambda) = (\lambda^2 - 1/\lambda^2)/(\lambda^2/q^2 - q^2/\lambda^2), \qquad (4.36)$$

we have similar properties for the right representations:

**Theorem 4.2** (Right $\mathscr{B}_-(\lambda)$ SOV-representations). *If $b_-(\lambda) \neq 0$ and (4.23)-(4.24) are satisfied, the states:*

$$|h_1, ..., h_N\rangle \equiv \frac{1}{N}\prod_{n=1}^{N}\prod_{k_n=h_n}^{p-2}\frac{\mathscr{D}_-(\xi_n^{(k_n+1)})}{\kappa_n^{(k_n+1)}A_-(1/\xi_n^{(k_n)})}|\bar{\Omega}\rangle, \qquad (4.37)$$

*with N the same coefficient as in (4.25), define a $\mathscr{B}_-(\lambda)$-eigenstates basis of $\mathscr{H}$:*

$$\mathscr{B}_-(\lambda)|\mathbf{h}\rangle = |\mathbf{h}\rangle B_{\mathbf{h}}(\lambda). \qquad (4.38)$$

*On the generic state $|\boldsymbol{h}\rangle$, the action of the remaining reflection algebra generators follows by:*

$$\mathscr{D}_-(\lambda)|\boldsymbol{h}\rangle = \sum_{a=1}^{2N} T_a^{-\varphi_a}|\boldsymbol{h}\rangle \frac{(\lambda^2/q - q/\lambda^2)(\lambda\zeta_a^{(h_a)} - 1/\zeta_a^{(h_a)}\lambda)}{((\zeta_a^{(h_a)})^2/q - q/(\zeta_a^{(h_a)})^2)((\zeta_a^{(h_a)})^2 - 1/(\zeta_a^{(h_a)})^2)} \prod_{\substack{b=1 \\ b \neq a\ modN}}^{N} \frac{\Lambda - X_b^{(h_b)}}{X_a^{(h_a)} - X_b^{(h_b)}}$$

$$\times D_-(\zeta_a^{(h_a)}) + |\boldsymbol{h}\rangle(-1)^N det_q M(1)\frac{(\lambda/q^{1/2} + q^{1/2}/\lambda)}{2} \prod_{b=1}^{N} \frac{\Lambda - X_b^{(h_b)}}{X - X_b^{(h_b)}}$$

$$+ (-1)^{N+1}|\boldsymbol{h}\rangle\frac{(\zeta_- + 1/\zeta_-)}{(\zeta_- - 1/\zeta_-)} det_q M(i)\frac{(\lambda/q^{1/2} - q^{1/2}/\lambda)}{2} \prod_{b=1}^{N} \frac{\Lambda - X_b^{(h_b)}}{X + X_b^{(h_b)}}, \tag{4.39}$$

*where:*

$$D_-(\lambda) = k(\lambda)A_-(q/\lambda), \quad T_a^{\pm}|h_1,...,h_a,...,h_N\rangle = |h_1,...,h_a \pm 1,...,h_N\rangle. \tag{4.40}$$

*Indeed, the representation of $\mathscr{A}_-(\lambda)$ follows from the identity (3.18) while $\mathscr{C}_-(\lambda)$ is given by the quantum determinant.*

*Proof.* The proof is given along the same steps used in the previous theorem, we just need to make the following remarks. First of all by using the identities (4.15) it follows that $\left|\bar{\Omega}\right\rangle$ is a $\mathscr{B}_-(\lambda)$-eigenstate with non-zero eigenvalue:

$$\mathscr{B}_-(\lambda)\left|\bar{\Omega}\right\rangle \equiv \left|\bar{\Omega}\right\rangle B_{p-1}(\lambda). \tag{4.41}$$

Now all we need are the following reflection algebra commutation relations:

$$\mathscr{B}_-(\lambda_1)\mathscr{D}_-(\lambda_2) = \frac{(\lambda_1 q/\lambda_2 - \lambda_2/(\lambda_1 q))(\lambda_1\lambda_2/q - q/(\lambda_1\lambda_2))}{(\lambda_1/\lambda_2 - \lambda_2/\lambda_1)(\lambda_1\lambda_2 - 1/(\lambda_1\lambda_2))}\mathscr{D}_-(\lambda_2)\mathscr{B}_-(\lambda_1)$$

$$- \frac{(q - 1/q)(\lambda_1\lambda_2/q - q/(\lambda_1\lambda_2))}{(\lambda_1/\lambda_2 - \lambda_2/\lambda_1)(\lambda_1\lambda_2 - 1/(\lambda_1\lambda_2))}\mathscr{D}_-(\lambda_1)\mathscr{B}_-(\lambda_2)$$

$$- \frac{q - 1/q}{(\lambda_1\lambda_2 - 1/(\lambda_1\lambda_2))}\mathscr{A}_-(\lambda_1)\mathscr{B}_-(\lambda_2). \tag{4.42}$$

By using them, the definition of the states $|\boldsymbol{h}\rangle$ and the quantum determinant, we get our interpolation formula for the right action of $\mathscr{D}_-(\lambda)$ on $|\boldsymbol{h}\rangle$. Let us remark that in fact, the chosen gauge for the coefficients of $\mathscr{D}_-(\lambda)$ is consistent with the quantum determinant condition as we have:

$$D_-(\zeta_a^{(h)}) = \kappa_a^{(h)} A_-(q/\zeta_a^{(h)}) \tag{4.43}$$

for any $h \in \{0,...,p-1\}$ and $a \in \{1,...,2N\}$ and

$$\frac{det_q \mathscr{U}_-(\xi_a^{(h+1/2)})}{((\xi_a^{(h+3/2)})^2 - 1/(\xi_a^{(h+3/2)})^2)} = D_-(\xi_a^{(h+1)})D_-(1/\xi_a^{(h)}) = A_-(\xi_a^{(h+1)})A_-(1/\xi_a^{(h)}), \tag{4.44}$$

since,

$$\kappa_a^{(h+1)}\kappa_{a+N}^{(h)} = 1 \tag{4.45}$$

for any $h \in \{0,...,p-1\}$ and $a \in \{1,...,N\}$. $\qquad\square$

## 4.3 Change of basis and SoV spectral decomposition of the identity

In this section we present the main properties of the $p^N \times p^N$ matrices $U^{(L)}$ and $U^{(R)}$ defining respectively the change of basis from the original left and right basis, formed by the $v_n$-eigenstates basis:

$$\underline{\langle h |} \equiv \otimes_{n=1}^N \langle h_n, n | \quad \text{and} \quad \underline{|h\rangle} \equiv \otimes_{n=1}^N |h_n, n\rangle, \tag{4.46}$$

to the left and right $\mathscr{B}_-$-eigenstates basis:

$$\langle h | = \underline{\langle h |} U^{(L)} = \sum_{i=1}^{p^N} U^{(L)}_{\varkappa(h),i} \langle \varkappa^{-1}(i) | \quad \text{and} \quad |h\rangle = U^{(R)} \underline{|h\rangle} = \sum_{i=1}^{p^N} U^{(R)}_{i,\varkappa(h)} |\varkappa^{-1}(i)\rangle, \tag{4.47}$$

where $\varkappa$ is the isomorphism between the sets $\{0, ..., p-1\}^N$ and $\{1, ..., p^N\}$ defined by:

$$\varkappa : h \in \{0, ..., p-1\}^N \to \varkappa(h) \equiv 1 + \sum_{a=1}^N p^{(a-1)} h_a \in \{1, ..., p^N\}. \tag{4.48}$$

From the diagonalizability of $\mathscr{B}_-(\lambda)$ it follows that $U^{(L)}$ and $U^{(R)}$ are invertible matrices for which it holds:

$$U^{(L)} \mathscr{B}_-(\lambda) = \Delta_{\mathscr{B}_-}(\lambda) U^{(L)}, \quad \mathscr{B}_-(\lambda) U^{(R)} = U^{(R)} \Delta_{\mathscr{B}_-}(\lambda), \tag{4.49}$$

where $\Delta_{\mathscr{B}_-}(\lambda)$ is the $p^N \times p^N$ diagonal matrix defined by:

$$\left( \Delta_{\mathscr{B}_-}(\lambda) \right)_{i,j} \equiv \delta_{i,j} B_{\varkappa^{-1}(i)}(\lambda) \quad \forall i, j \in \{1, ..., p^N\}. \tag{4.50}$$

We can prove that it holds:

**Proposition 4.1.** *The $p^N \times p^N$ matrix $M \equiv U^{(L)} U^{(R)}$ consisting of scalar products of left and right $\mathscr{B}_-$-eigenstates is diagonal and it is characterized by the following diagonal entries:*

$$M_{\varkappa(h)\varkappa(h)} = \langle h | h \rangle = \prod_{1 \le b < a \le N} \frac{1}{X_a^{(h_a)} - X_b^{(h_b)}}. \tag{4.51}$$

*Proof.* Note that the action of a left $\mathscr{B}_-$-eigenstate on a right $\mathscr{B}_-$-eigenstate is zero, for two different $\mathscr{B}_-$-eigenvalues. This implies that the matrix $M$ is diagonal; then to compute its diagonal elements we compute the matrix elements

$$\theta_a \equiv \langle h_1, ..., h_a, ..., h_N | \mathscr{A}_-(\xi_a^{(h_a+1)}) | h_1, ..., h_a+1, ..., h_N \rangle \quad \text{where} \quad a \in \{1, ..., N\}. \tag{4.52}$$

Using the left action of the operator $\mathscr{A}_-(\xi_a^{(h_a+1)})$ we get:

$$\theta_a = \frac{(q-1/q) A_-(1/\xi_a^{(h_a)})}{((\xi_a^{(h_a)})^2 - 1/(\xi_a^{(h_a)})^2)} \prod_{\substack{b=1 \\ b \ne a}}^N \frac{X_a^{(h_a+1)} - X_b^{(h_b)}}{X_a^{(h_a)} - X_b^{(h_b)}}$$

$$\times \langle h_1, ..., h_a+1, ..., h_N | h_1, ..., h_a+1, ..., h_N \rangle \tag{4.53}$$

while using the decomposition (3.18) and the fact that:

$$\langle h_1, ..., h_a, ..., h_N | \mathscr{D}_-(1/\xi_a^{(h_a+1)}) | h_1, ..., h_a+1, ..., h_N \rangle = 0 \tag{4.54}$$

it holds:

$$\theta_a = \frac{k_a^{(h_a+1)}(q-1/q) A_-(1/\xi_a^{(h_a)})}{((\xi_a^{(h_a+1)})^2 - 1/(\xi_a^{(h_a+1)})^2)} \langle h_1, ..., h_a, ..., h_N | h_1, ..., h_a, ..., h_N \rangle, \tag{4.55}$$

and so:

$$\theta_a = \frac{(q-1/q)A_-(1/\xi_a^{(h_a)})}{((\xi_a^{(h_a)})^2 - 1/(\xi_a^{(h_a)})^2)}\langle h_1,...,h_a,...,h_N|h_1,...,h_a,...,h_N\rangle. \tag{4.56}$$

These results lead to the identity:

$$\frac{\langle h_1,...,h_a+1,...,h_N|h_1,...,h_a+1,...,h_N\rangle}{\langle h_1,...,h_a,...,h_N|h_1,...,h_a,...,h_N\rangle} = \prod_{\substack{b=1\\b\neq a}}^{N}\frac{X_a^{(h_a)}-X_b^{(h_b)}}{X_a^{(h_a+1)}-X_b^{(h_b)}}, \tag{4.57}$$

from which one can prove:

$$\frac{\langle h_1,...,h_N|h_1,...,h_N\rangle}{\langle p-1,...,p-1|p-1,...,p-1\rangle} = \prod_{1\leq b<a\leq N}\frac{X_a^{(p-1)}-X_b^{(p-1)}}{X_a^{(h_a)}-X_b^{(h_b)}}. \tag{4.58}$$

This proves the proposition being

$$\langle p-1,...,p-1|p-1,...,p-1\rangle = \prod_{1\leq b<a\leq N}\frac{1}{X_a^{(p-1)}-X_b^{(p-1)}}, \tag{4.59}$$

using the following choice of the normalization:

$$N = \left(\prod_{1\leq b<a\leq N}\left(X_a^{(p-1)}-X_b^{(p-1)}\right)\langle\Omega|\prod_{n=1}^{N}\prod_{k_n=0}^{p-2}\frac{\mathscr{A}_-(1/\xi_n^{(k_n)})}{A_-(1/\xi_n^{(k_n)})}|\bar{\Omega}\rangle\right)^{1/2}. \tag{4.60}$$

$\square$

The previous theorem implies the following spectral decomposition of the identity $\mathbb{I}$ in the SoV basis:

$$\mathbb{I} \equiv \sum_{h_1,...,h_N=0}^{p-1}\prod_{1\leq b<a\leq N}(X_a^{(h_a)}-X_a^{(h_a)})|h_1,...,h_N\rangle\langle h_1,...,h_N|. \tag{4.61}$$

## 4.4 Separate states and their scalar products

Let us introduce a class of left and right states, the so-called separate states, characterized by the following type of decompositions in the left and right SoV-basis:

$$\langle\alpha| = \sum_{h_1,...,h_N=0}^{p-1}\prod_{a=1}^{N}\alpha_a^{(h_a)}\prod_{1\leq b<a\leq N}(X_a^{(h_a)}-X_b^{(h_b)})\langle h_1,...,h_N|, \tag{4.62}$$

$$|\beta\rangle = \sum_{h_1,...,h_N=0}^{p-1}\prod_{a=1}^{N}\beta_a^{(h_a)}\prod_{1\leq b<a\leq N}(X_a^{(h_a)}-X_b^{(h_b)})|h_1,...,h_N\rangle, \tag{4.63}$$

where the coefficients $\alpha_a^{(h_a)}$ and $\beta_a^{(h_a)}$ are arbitrary complex numbers, meaning that the coefficients of these separate states have a factorized form in this basis. These separate states are interesting at least for two reasons : they admit simple determinant scalar products, as it will be shown in the next proposition, and the eigenstates of the transfer matrix are special separate states, as we will show in the next section.

**Proposition 4.2.** *Let us take an arbitrary separate left state $\langle\alpha|$ (separate covector) and an arbitrary separate right state $|\beta\rangle$ (separate vector) then it holds:*

$$\langle\alpha|\beta\rangle = det_N||\mathscr{M}_{a,b}^{(\alpha,\beta)}|| \quad with \quad \mathscr{M}_{a,b}^{(\alpha,\beta)} \equiv \sum_{h=0}^{p-1}\alpha_a^{(h)}\beta_a^{(h)}(X_a^{(h)})^{(b-1)}. \tag{4.64}$$

*Proof.* The proof follows the same method as in [97]. The formula (4.51) implies, using the representation of the states $\langle\alpha|$ and $|\beta\rangle$, the following:

$$\langle\alpha|\beta\rangle = \sum_{h_1,...,h_N=0}^{p-1} V(X_1^{(h_1)},...,X_N^{(h_N)})\prod_{a=1}^{N}\alpha_a^{(h_a)}\beta_a^{(h_a)}, \tag{4.65}$$

where we have denoted by $V(x_1,...,x_N) \equiv \prod_{1\leq b<a\leq N}(x_a-x_b)$ the Vandermonde determinant. Finally, using the multilinearity of the determinant we get our result. $\qquad\square$

# 5 $\mathscr{T}$-spectrum characterization in the SoV basis

In this section we present the complete characterization of the spectrum of the transfer matrix $\mathscr{T}(\lambda)$ associated to the cyclic representations of the 6-vertex reflection algebra. We first present some preliminary properties satisfied by all the eigenvalue functions of the transfer matrix $\mathscr{T}(\lambda)$:

**Lemma 5.1.** *Denote by $\Sigma_{\mathscr{T}}$ the transfer matrix spectrum, then any $\tau(\lambda) \in \Sigma_{\mathscr{T}}$ is an even function of $\lambda$ invariant under the transformation $\lambda \to 1/\lambda$ which admits the following interpolation formula:*

$$\tau(\lambda) = \sum_{a=1}^{N}\frac{\Lambda^2-X^2}{(X_a^{(0)})^2-X^2}\prod_{\substack{b=1\\b\neq a}}^{N}\frac{\Lambda-X_b^{(0)}}{X_a^{(0)}-X_b^{(0)}}\tau(\zeta_a^{(0)})+(-1)^N\frac{(\Lambda+X)}{2}\prod_{b=1}^{N}\frac{\Lambda-X_b^{(0)}}{X-X_b^{(0)}}det_qM(1)$$

$$-(-1)^N\frac{(\Lambda-X)}{2}\prod_{b=1}^{N}\frac{\Lambda-X_b^{(0)}}{X+X_b^{(0)}}\frac{(\zeta_++1/\zeta_+)}{(\zeta_+-1/\zeta_+)}\frac{(\zeta_-+1/\zeta_-)}{(\zeta_--1/\zeta_-)}det_qM(i)$$

$$+(\Lambda^2-X^2)\tau_\infty\prod_{b=1}^{N}(\Lambda-X_b^{(0)}), \tag{5.1}$$

*where:*

$$\tau_\infty \equiv \frac{\kappa_+\kappa_-(e^{\tau_+-\tau_-}\prod_{b=1}^{N}\delta_b\gamma_b+e^{\tau_--\tau_+}\prod_{b=1}^{N}\alpha_b\beta_b)}{(\zeta_+-1/\zeta_+)(\zeta_--1/\zeta_-)}. \tag{5.2}$$

*Proof.* In the previous section, we have shown that the transformations $\lambda \to -\lambda$ and $\lambda \to 1/\lambda$ are symmetries of the transfer matrix $\mathscr{T}(\lambda)$ so if $\tau(\lambda) \in \Sigma_{\mathscr{T}}$ then $\tau(\lambda)$ is left unchanged under these transformations. Moreover, the asymptotic of the transfer matrix can be easily derived by direct computations, it is central and it holds:

$$\tau_\infty = \lim_{\log\lambda\to\pm\infty}\lambda^{\mp2(N+2)}\mathscr{T}(\lambda). \tag{5.3}$$

The identities (4.33) imply after some simple computation that the transfer matrix is central in $q^{\pm1/2}$ and $iq^{\pm1/2}$ and that it holds:

$$\mathscr{T}(q^{\pm1/2}) = (-1)^N X det_q M(1), \quad \mathscr{T}(iq^{\pm1/2}) = (-1)^N X\frac{(\zeta_++1/\zeta_+)}{(\zeta_+-1/\zeta_+)}\frac{(\zeta_-+1/\zeta_-)}{(\zeta_--1/\zeta_-)}det_q M(i). \tag{5.4}$$

The known functional form of $\mathscr{T}(\lambda)$ with regards to $\lambda$ together with this identities imply the interpolation formula in the lemma. $\qquad\square$

The previous lemma defines the set of polynomials to which belong the transfer matrix eigenvalues; in order to completely characterize the eigenvalues we introduce now the following one-parameter family $D_\tau(\lambda)$ of $p \times p$ matrices:

$$D_\tau(\lambda) \equiv \begin{pmatrix} \tau(\lambda) & -\text{A}(1/\lambda) & 0 & \cdots & 0 & -\text{A}(\lambda) \\ -\text{A}(q\lambda) & \tau(q\lambda) & -\text{A}(1/(q\lambda)) & 0 & \cdots & 0 \\ 0 & \ddots & & & & \vdots \\ \vdots & & \cdots & & & \vdots \\ \vdots & & & \cdots & & \vdots \\ \vdots & & & & \ddots & 0 \\ 0 & \cdots & 0 & -\text{A}(q^{2l-1}\lambda) & \tau(q^{2l-1}\lambda) & -\text{A}(1/(q^{2l-1}\lambda)) \\ -\text{A}(1/(q^{2l}\lambda)) & 0 & \cdots & 0 & -\text{A}(q^{2l}\lambda) & \tau(q^{2l}\lambda) \end{pmatrix},$$

(5.5)

where for now $\tau(\lambda)$ is a generic function and we have defined:

$$\text{A}(\lambda) = \text{a}_+(\lambda)\text{A}_-(\lambda). \tag{5.6}$$

Note that the coefficient $\text{A}(\lambda)$ satisfies the quantum determinant condition:

$$\text{A}(\lambda q^{1/2})\text{A}(q^{1/2}/\lambda) = \frac{\text{a}_+(\lambda q^{1/2})\text{a}_+(q^{1/2}/\lambda)\det_q \mathscr{U}_-(\lambda)}{(\lambda/q)^2 - (q/\lambda)^2}. \tag{5.7}$$

**Lemma 5.2.** *Let $\tau(\lambda)$ be a function of $\lambda$ invariant under the transformation $\lambda \to 1/\lambda$ then $\det_p D_\tau(\lambda)$ is a function of $\lambda^p$ invariant under the transformation $\lambda^p \to 1/\lambda^p$.*

*Proof.* Let us observe that for the invariance of the function $\tau(\lambda)$ under $\lambda \to 1/\lambda$, we have that:

$$D_\tau(1/\lambda) = O_C O_R(D_\tau(\lambda)), \tag{5.8}$$

where we have denoted by $O_R$ the operation on a $p \times p$ matrix which exchanges the couple of rows $p - i$ with $i + 2$ for any $i \in \{0, ..., (p-3)/2\}$, similarly $O_C$ is the operation on a $p \times p$ matrix which exchanges the couple of columns $p - i$ with $i + 2$ for any $i \in \{0, ..., (p-3)/2\}$. It is then trivial to see that:

$$\det_p D_\tau(1/\lambda) = \det_p D_\tau(\lambda). \tag{5.9}$$

Let us now observe that:

$$D_\tau(\lambda q) = C_{p \to 1} R_{p \to 1}(D_\tau(\lambda)), \tag{5.10}$$

where $R_{p \to 1}$ is the operation on a $p \times p$ matrix which move the last row in the first row leaving the order of the others unchanged and similarly $C_{p \to 1}$ is the operation on a $p \times p$ matrix which move the last column in the first column leaving the order of the others unchanged. This clearly implies that:

$$\det_p D_\tau(q\lambda) = \det_p D_\tau(\lambda), \tag{5.11}$$

which completes the proof of the lemma. $\qquad\square$

As mentioned before, we will restrict our attention to the special boundary condition $b_+(\lambda) = 0$, for which the SoV-basis coincides with the $\mathscr{B}_-$-eigenstates basis. That is we consider the spectral problem of the transfer matrix:

$$\mathscr{T}(\lambda) \equiv \mathscr{T}_\backslash(\lambda) + c_+(\lambda)\,\mathscr{B}_-(\lambda), \tag{5.12}$$

under the following conditions on the boundary parameters:

$$b_+(\lambda) = 0 \quad \text{and} \quad b_-(\lambda) \neq 0. \tag{5.13}$$

Note that the condition $b_+(\lambda) = 0$, keeping instead if desired a $c_+(\lambda) \neq 0$, can be simply realized by the following renormalization of the boundary parameters $\kappa_+ = e^{-\gamma}\kappa'_+$ and $e^{\tau_+} = e^{\tau'_+-\gamma}$ by sending $\gamma \to +\infty$. Under this limit the asymptotic of the transfer matrix reads:

$$\tau_\infty = \frac{(-1)^N \kappa'_+ \kappa_- e^{\tau_- - \tau'_+} \prod_{b=1}^{N} \alpha_b \beta_b}{(\zeta_+ - 1/\zeta_+)(\zeta_- - 1/\zeta_-)}. \tag{5.14}$$

In the following we will suppress the unnecessary prime in $\kappa_+$ and $\tau_+$.

**Theorem 5.1.** *If the conditions:*

$$\mu_{n,+}^2 \neq q^{-2h}\zeta_+^{\pm 2}, \quad \forall h \in \{1, ..., p-1\}, \ n \in \{1, ..., N\} \tag{5.15}$$

*and* (4.23)-(4.24) *are satisfied, then $\mathcal{T}(\lambda)$ has simple spectrum and $\Sigma_{\mathcal{T}}$ coincides with the set of polynomials $\tau(\lambda)$ of the form* (5.1) *with* (5.14) *which satisfy the following discrete system of equations:*

$$det D_\tau(\zeta_a^{(0)}) = 0, \ \forall a \in \{1, ..., N\}. \tag{5.16}$$

*I) The right $\mathcal{T}$-eigenstate corresponding to $\tau(\lambda) \in \Sigma_{\mathcal{T}}$ is defined by the following decomposition in the right SoV-basis:*

$$|\tau\rangle = \sum_{h_1,...,h_N=0}^{p-1} \prod_{a=1}^{N} Q_{\tau,a}^{(h_a)} \prod_{1 \leq b < a \leq N} (X_a^{(h_a)} - X_b^{(h_b)})|h_1, ..., h_N\rangle, \tag{5.17}$$

*where the $Q_{\tau,a}^{(h_a)}$ are the unique nontrivial solution up to normalization of the linear homogeneous system:*

$$D_\tau(\zeta_a^{(0)}) \begin{pmatrix} Q_{\tau,a}^{(0)} \\ \vdots \\ Q_{\tau,a}^{(p-1)} \end{pmatrix} = \begin{pmatrix} 0 \\ \vdots \\ 0 \end{pmatrix}. \tag{5.18}$$

*II) The left $\mathcal{T}$-eigenstate corresponding to $\tau(\lambda) \in \Sigma_{\mathcal{T}}$ is defined by the following decomposition in the left SoV-basis:*

$$\langle\tau| = \sum_{h_1,...,h_N=0}^{p-1} \prod_{a=1}^{N} \hat{Q}_{\tau,a}^{(h_a)} \prod_{1 \leq b < a \leq N} (X_a^{(h_a)} - X_b^{(h_b)})\langle h_1, ..., h_N|, \tag{5.19}$$

*where the $\hat{Q}_{\tau,a}^{(h_a)}$ are the unique nontrivial solution up to normalization of the linear homogeneous system:*

$$\begin{pmatrix} \hat{Q}_{\tau,a}^{(0)} & \cdots & \hat{Q}_{\tau,a}^{(p-1)} \end{pmatrix} \left(\hat{D}_\tau(\zeta_a^{(0)})\right)^{t_0} = \begin{pmatrix} 0 & \cdots & 0 \end{pmatrix}, \tag{5.20}$$

*and $\hat{D}_\tau(\lambda)$ is the family of $p \times p$ matrices defined substituting in $D_\tau(\lambda)$ the coefficient $A(\lambda)$ with*

$$D(\lambda) = d_+(\lambda)D_-(\lambda). \tag{5.21}$$

*Finally, using ideas from [97], let us note that if $\tau(\lambda) \neq \tau'(\lambda) \in \Sigma_{\mathcal{T}}$:*

$$\sum_{b=1}^{N} \mathcal{M}_{a,b}^{(\tau,\tau')} x_b^{(\tau,\tau')} = 0 \quad \forall a \in \{1, ..., N\}, \tag{5.22}$$

*where the $x_b^{(\tau,\tau')}$ are defined by:*

$$\tau(\lambda) - \tau'(\lambda) \equiv (\Lambda^2 - X^2) \sum_{b=1}^{N} x_b^{(\tau,\tau')} \Lambda^{b-1}, \tag{5.23}$$

*which in particular implies that the action of $\langle\tau|$ on $|\tau'\rangle$ is zero.*

*Proof.* The spectrum (eigenvalues and eigenstates) of the transfer matrix $\mathcal{T}(\lambda)$ in the SoV-basis is characterized by the following discrete system of equations:

$$\tau(\xi_n^{(h_n)})\Psi_\tau(\boldsymbol{h}) = \mathrm{A}(\xi_n^{(h_n)})\Psi_\tau(\mathsf{T}_n^-(\boldsymbol{h})) + \mathrm{A}(1/\xi_n^{(h_n)})\Psi_\tau(\mathsf{T}_n^+(\boldsymbol{h})), \tag{5.24}$$

for any $n \in \{1,...,\mathsf{N}\}$ and $\boldsymbol{h} \in \{0,...,p-1\}^\mathsf{N}$, i.e. a system of $p^\mathsf{N}$ Baxter-like equations in the *wave-functions*:

$$\Psi_\tau(\boldsymbol{h}) \equiv \langle h_1,...,h_\mathsf{N}|\tau\rangle, \tag{5.25}$$

of the $\mathcal{T}$-eigenstate $|\tau\rangle$ associated to $\tau(\lambda) \in \Sigma_{\mathcal{T}}$, where:

$$\mathsf{T}_n^\pm(\boldsymbol{h}) \equiv (h_1,\ldots,h_n \pm 1,\ldots,h_\mathsf{N}). \tag{5.26}$$

This system admits the following equivalent representation as $\mathsf{N}$ linear systems of homogeneous equations:

$$D_\tau(\xi_n^{(0)})\begin{pmatrix} \Psi_\tau(h_1,...,h_n=0,...,h_\mathsf{N}) \\ \Psi_\tau(h_1,...,h_n=1,...,h_\mathsf{N}) \\ \vdots \\ \Psi_\tau(h_1,...,h_n=p-1,...,h_\mathsf{N}) \end{pmatrix} = \begin{pmatrix} 0 \\ 0 \\ \vdots \\ 0 \end{pmatrix}, \tag{5.27}$$

for any $n \in \{1,...,\mathsf{N}\}$ and for any $h_{m\neq n}$ in $\{0,...,p-1\}$. Then the condition $\tau(\lambda) \in \Sigma_{\mathcal{T}}$ implies the compatibility equations for these linear systems, i.e. it must hold:

$$\det D_\tau(\xi_a^{(0)}) = 0, \ \forall a \in \{1,...,\mathsf{N}\} \tag{5.28}$$

Note that for the previous lemma this condition is verified also in the points $\zeta_a^{(0)} = 1/\xi_{a-\mathsf{N}}^{(0)}$ for any $a \in \{\mathsf{N}+1,...,2\mathsf{N}\}$. The rank of the matrices in (5.27) is $p-1$ being

$$\mathrm{A}(\xi_n^{(h)}) \neq 0, \ \mathrm{A}(1/\xi_n^{(h-1)}) \neq 0 \ \forall h \in \{1,...,p-1\}, \ n \in \{1,...,\mathsf{N}\}, \tag{5.29}$$

for the conditions (4.23), (4.24) and (5.15). Then (up to an overall normalization) the solution is unique and independent from the $h_{m\neq n} \in \{0,...,p-1\}$ for any $n \in \{1,...,\mathsf{N}\}$. So fixing $\tau(\lambda) \in \Sigma_{\mathcal{T}}$ there exists (up to normalization) one and only one corresponding $\mathcal{T}$-eigenstate $|\tau\rangle$ with coefficients of the factorized form given in (5.17)-(5.18); i.e. the $\mathcal{T}$-spectrum is simple.

Vice versa, if $\tau(\lambda)$ is in the set of functions (5.1) and satisfies (5.16), then the state $|\tau\rangle$ defined by (5.17)-(5.18) satisfies:

$$\langle h_1,...,h_\mathsf{N}|\mathcal{T}(\zeta_n^{(h_n)})|\tau\rangle = \tau(\zeta_n^{(h_n)})\langle h_1,...,h_\mathsf{N}|\tau\rangle \ \forall n \in \{1,...,\mathsf{N}\} \tag{5.30}$$

for any $\mathcal{B}_-$-eigenstate $\langle h_1,...,h_\mathsf{N}|$ and this implies:

$$\langle h_1,...,h_\mathsf{N}|\mathcal{T}(\lambda)|\tau\rangle = \tau(\lambda)\langle h_1,...,h_\mathsf{N}|\tau\rangle \ \ \forall \lambda \in \mathbb{C}, \tag{5.31}$$

i.e. $\tau(\lambda) \in \Sigma_{\mathcal{T}}$ and $|\tau\rangle$ is the corresponding $\mathcal{T}$-eigenstate.

For the left $\mathcal{T}$-eigenstates the proof follows as above, we just remark in this case that the matrix elements:

$$\langle\tau|\mathcal{T}(\zeta_n^{(h_n)})|h_1,...,h_\mathsf{N}\rangle, \tag{5.32}$$

are computed in the right $\mathcal{B}_-$-representation:

$$\tau(\zeta_n^{(h_n)})\hat{\Psi}_\tau(\boldsymbol{h}) = \mathrm{D}(\zeta_n^{(h_n)})\hat{\Psi}_\tau(\mathsf{T}_n^-(\boldsymbol{h})) + \mathrm{D}(1/\zeta_n^{(h_n)})\hat{\Psi}_\tau(\mathsf{T}_n^+(\boldsymbol{h})), \ \ \forall n \in \{1,...,\mathsf{N}\} \tag{5.33}$$

where:

$$\hat{\Psi}_\tau(\boldsymbol{h}) \equiv \langle\tau|h_1,...,h_\mathsf{N}\rangle, \ \ \mathrm{D}(\lambda) \equiv \mathrm{d}_+(\lambda)\mathrm{D}_-(\lambda). \tag{5.34}$$

It is simple to observe that it holds:

$$\det D_\tau(\lambda) = \det \hat{D}_\tau(\lambda), \tag{5.35}$$

as a consequence of the following identities:

$$\text{D}(\lambda) = \alpha(\lambda)\text{A}(q/\lambda) \tag{5.36}$$

and:

$$\alpha(1/\lambda)\alpha(q\lambda) = 1, \quad \prod_{a=0}^{p-1} \alpha(\lambda q^a) = 1 \tag{5.37}$$

where:

$$\alpha(\lambda) = \frac{s(\lambda)}{s(q/\lambda)}k(\lambda), \quad s(\lambda) = \frac{\lambda^2 q - 1/(q\lambda^2)}{\lambda^2 - 1/\lambda^2}. \tag{5.38}$$

Finally, the identity (5.22) is quite general in the SoV framework and can be proven also in our case of cyclic representations of the 6-vertex reflection algebra by following the same proof first given in the case of a periodic lattice [97]. □

For next applications it is interesting to show that we can obtain the coefficients of a left transfer matrix eigenstates in terms of those of the right one by introducing a recursion formula that produces both coefficients in terms of the transfer matrix eigenvalues. The following lemma holds:

**Lemma 5.3.** *Let $\tau(\lambda) \in \Sigma_{\mathcal{T}}$ then we have:*

$$\frac{\hat{Q}_{\tau,a}^{(h)}}{\hat{Q}_{\tau,a}^{(h-1)}} = \frac{\text{A}(1/\zeta_a^{(h-1)})}{\text{D}(1/\zeta_a^{(h-1)})} \frac{Q_{\tau,a}^{(h)}}{Q_{\tau,a}^{(h-1)}}, \tag{5.39}$$

*being:*

$$\frac{\hat{Q}_{\tau,a}^{(h)}}{\hat{Q}_{\tau,a}^{(0)}} = \frac{t_{\tau,a}^{(h)}}{\prod_{b=0}^{h-1} \text{D}(1/\zeta_a^{(b)})}, \quad \frac{Q_{\tau,a}^{(h)}}{Q_{\tau,a}^{(0)}} = \frac{t_{\tau,a}^{(h)}}{\prod_{b=0}^{h-1} \text{A}(1/\zeta_a^{(b)})}, \tag{5.40}$$

*where the $t_{\tau,a}^{(h)}$ are defined by the following recursion formula:*

$$t_{\tau,a}^{(h)} = \tau(\zeta_a^{(h-1)})t_{\tau,a}^{(h-1)} - \frac{det_q K_+(\xi_a^{(h-3/2)})det_q \mathcal{U}_-(\xi_a^{(h-3/2)})}{(\xi_a^{(h-1/2)})^2 - 1/(\xi_a^{(h-1/2)})^2}t_{\tau,a}^{(h-2)} \, for \, h \in \{1,...,p-1\} \tag{5.41}$$

*with the following initial conditions $t_{\tau,a}^{(-1)} = 0$, $t_{\tau,a}^{(0)} = 1$.*

*Proof.* We just need to prove the last two identities in this lemma as the first ones are simple consequences of them. Let us prove the second identity in (5.40). For $h = 1$ we have:

$$\frac{Q_{\tau,a}^{(1)}}{Q_{\tau,a}^{(0)}} = \frac{\tau(\zeta_a^{(0)})}{\text{A}(1/\zeta_a^{(0)})}, \tag{5.42}$$

by the SoV Baxter's like equation and the condition:

$$\text{A}(\zeta_a^{(0)}) = 0, \quad \text{A}(1/\zeta_a^{(0)}) \neq 0. \tag{5.43}$$

This means that the second formula in (5.40) and (5.41) are both satisfied for $h = 1$ once the initial conditions $t_{\tau,a}^{(-1)} = 0$, $t_{\tau,a}^{(0)} = 1$ are imposed. So let us assume that these two identities

are satisfied for any $h \in \{1, ..., x\}$ and let us prove it for $h = x + 1 \le p - 1$. We have that by the SoV equations it holds:

$$\frac{Q_{\tau,a}^{(x+1)}}{Q_{\tau,a}^{(x-1)}} = \frac{\tau(\zeta_a^{(x)})}{A(1/\zeta_a^{(x)})} \frac{Q_{\tau,a}^{(x)}}{Q_{\tau,a}^{(x-1)}} - \frac{A(\zeta_a^{(x)})}{A(1/\zeta_a^{(x)})} \tag{5.44}$$

and so:

$$\frac{Q_{\tau,a}^{(x+1)}}{Q_{\tau,a}^{(0)}} = \frac{\tau(\zeta_a^{(x)})}{A(1/\zeta_a^{(x)})} \frac{Q_{\tau,a}^{(x)}}{Q_{\tau,a}^{(0)}} - \frac{A(\zeta_a^{(x)})}{A(1/\zeta_a^{(x)})} \frac{Q_{\tau,a}^{(x-1)}}{Q_{\tau,a}^{(0)}} \tag{5.45}$$

which by using the formula (5.40) for $h = x - 1$ and $h = x$ reads:

$$\frac{Q_{\tau,a}^{(x+1)}}{Q_{\tau,a}^{(0)}} = \frac{\tau(\zeta_a^{(x)})t_{\tau,a}^{(x)} - A(\zeta_a^{(x)})A(1/\zeta_a^{(x-1)})t_{\tau,a}^{(x-1)}}{\prod_{b=0}^{x} A(1/\zeta_a^{(b)})}, \tag{5.46}$$

which just proves the formulae (5.40) and (5.41) for $h = x + 1$ once we recall that:

$$A(\zeta_a^{(x)})A(1/\zeta_a^{(x-1)}) = \frac{\det_q K_{a,+}(\xi_a^{(x-1/2)})\det_q \mathscr{U}_{a,-}(\xi_a^{(x-1/2)})}{(\xi_a^{(h+1/2)})^2 - 1/(\xi_a^{(h+1/2)})^2}. \tag{5.47}$$

The proof of the first identity in (5.40) can be done in the same way, we have just to use now that:

$$D(\zeta_a^{(x)})D(1/\zeta_a^{(x-1)}) = \frac{\det_q K_{a,+}(\xi_a^{(x-1/2)})\det_q \mathscr{U}_{a,-}(\xi_a^{(x-1/2)})}{(\xi_a^{(h+1/2)})^2 - 1/(\xi_a^{(h+1/2)})^2}. \tag{5.48}$$

$\square$

# 6 Functional equation characterizing the $\mathscr{T}$-spectrum

In this section we assume that at any quantum site the following constraints are satisfied:

$$b_n^p + a_n^p = 0, \quad \forall n \in \{1, ..., N\}, \tag{6.1}$$

$$c_n^p + d_n^p = 0, \quad \forall n \in \{1, ..., N\}. \tag{6.2}$$

Under these conditions we can explicitly construct the left and right basis which diagonalize the one-parameter family of commuting operators $\mathscr{B}_-(\lambda)$ and $\mathscr{C}_-(\lambda)$ associated to the most general $K_-(\lambda)$ matrix. In the previous section we have done this construction for the $\mathscr{B}_-(\lambda)$ eigenstates basis, clearly we can do a similar construction also for $\mathscr{C}_-(\lambda)$. In the previous sections we have explained how the spectral problem of the transfer matrix $\mathscr{T}(\lambda)$ can be characterized by SoV in the $\mathscr{B}_-(\lambda)$ eigenstates basis when $b_+(\lambda) = 0$ keeping instead if desired a $c_+(\lambda) \ne 0$, by the same approach we can characterize the $\mathscr{T}(\lambda)$-spectrum by SoV in the $\mathscr{C}_-(\lambda)$ eigenstates basis when $c_+(\lambda) = 0$ keeping instead if desired a $b_+(\lambda) \ne 0$.

Here, we show that the SoV characterization of the $\mathscr{T}(\lambda)$-spectrum can be reformulated by functional equations. Before this, let us just observe that the central value of the asymptotic of $\mathscr{T}(\lambda)$ is given by:

$$\tau_\infty = \frac{(-1)^N \kappa_+ \kappa_- e^{\epsilon(\tau_- - \tau_+)} \prod_{b=1}^{N} \alpha_b \beta_b}{(\zeta_+ - 1/\zeta_+)(\zeta_- - 1/\zeta_-)}, \tag{6.3}$$

where $\epsilon = +1$ for $b_+(\lambda) = 0$ (keeping instead a $c_+(\lambda) \ne 0$) and $\epsilon = -1$ for $c_+(\lambda) = 0$ (keeping instead a $b_+(\lambda) \ne 0$). Let us moreover introduce the following notations:

$$\overline{A}(\lambda) = \lambda q^{-1/2} A(\lambda), \tag{6.4}$$

$$\overline{A}_\infty = \lim_{\lambda \to +\infty} \lambda^{-2(N+2)} \overline{A}(\lambda) = \frac{(-1)^N \alpha_- \beta_- \zeta_+ \kappa_- \prod_{n=1}^{N} b_n c_n}{q^{1+N}(\zeta_+ - 1/\zeta_+)(\zeta_- - 1/\zeta_-)}, \tag{6.5}$$

and:

$$F(\lambda) = \prod_{b=1}^{2N} \left( \frac{\lambda^p}{\left(\zeta_b^{(0)}\right)^p} - \frac{\left(\zeta_b^{(0)}\right)^p}{\lambda^p} \right),$$ (6.6)

then the following results hold:

**Proposition 6.1.** *If the conditions* (4.23), (4.24) *and* (5.15) *are satisfied and:*

$$b_n = -q^{2j_n-1}a_n, \quad d_n = -q^{2j_n-1}c_n,$$ (6.7)

*then $\mathscr{T}(\lambda)$ has simple spectrum and $\tau(\lambda)$ of the form* (5.1) *with* (5.14) *is an element of $\Sigma_{\mathscr{T}}$ if and only if $det_p \bar{D}_\tau(\lambda)$ is a Laurent polynomial of degree $N + 2$ in the variable:*

$$Z = \lambda^{2p} + \frac{1}{\lambda^{2p}}$$ (6.8)

*which satisfies the following functional equation:*

$$det_p \bar{D}_\tau(\lambda) - F(\lambda) \sum_{a=0}^{1} det_p \bar{D}_\tau(i^a q^{1/2}) \frac{(\lambda^p + (-1)^a /\lambda^p)^2}{4(-1)^a F(i^a q^{1/2})} = \left(\tau_\infty^p - \bar{A}_\infty^p\right) F(\lambda) \left(\lambda^{2p} - \frac{1}{\lambda^{2p}}\right)^2.$$ (6.9)

*Here $\bar{D}_\tau(\lambda)$ is obtained from $D_\tau(\lambda)$ by substituting in it $A(\lambda)$ with $\bar{A}(\lambda)$.*

*Proof.* Let us observe that $det_p \bar{D}_\tau(\lambda)$ is an even function of $\lambda$ as a consequence of the parity of $\tau(\lambda)$ and $\bar{A}(\lambda)$ moreover following the same steps of Lemma 5.2 we can prove that $det_p \bar{D}_\tau(\lambda)$ is invariant under the transformations $\lambda \to 1/\lambda$ and $\lambda \to q\lambda$, so that $det_p \bar{D}_\tau(\lambda)$ is indeed a function of $Z = \lambda^{2p} + 1/\lambda^{2p}$. Let us now observe that in the points $\lambda = q^{1/2}$ and $\lambda = iq^{1/2}$, the central row of the matrix $\bar{D}_\tau(\lambda)$ has two elements which are divergent as proportional to $\bar{A}(\pm q^{p/2})$ and $\bar{A}(\pm iq^{p/2})$, respectively. In the following we prove that:

$$det_p \bar{D}_\tau(q^{1/2}) \text{ and } det_p \bar{D}_\tau(iq^{1/2}) \text{ are finite},$$ (6.10)

if we ask that the function $\tau(\lambda)$ has the functional form (5.1). Let us first introduce the notations:

$$\bar{x}(\lambda) = \left(\lambda^2 - \frac{1}{\lambda^2}\right) \bar{A}(\lambda),$$ (6.11)

and using it let us expand the determinant:

$$det_p \bar{D}_\tau(\lambda q^{1/2}) = \tau(\lambda) det_{p-1} \bar{D}_{\tau,(p+1)/2,(p+1)/2}(\lambda q^{1/2}) + \frac{\bar{x}(\lambda) det_{p-1} \bar{D}_{\tau,(p+1)/2,(p+1)/2-1}(\lambda q^{1/2})}{\lambda^2 - 1/\lambda^2}$$
$$- \frac{\bar{x}(1/\lambda) det_{p-1} \bar{D}_{\tau,(p+1)/2,(p+1)/2+1}(\lambda q^{1/2})}{\lambda^2 - 1/\lambda^2},$$ (6.12)

with regards to the central row. Here, we have denoted with $\bar{D}_{\tau,i,j}(\lambda)$ the $(p-1) \times (p-1)$ matrix defined by removing the row $i$ and the column $j$ to the matrix $\bar{D}_\tau(\lambda)$. The following identity holds:

$$det_{p-1} \bar{D}_{\tau,(p+1)/2,(p+1)/2+1}(\lambda q^{1/2}) = det_{p-1} \bar{D}_{\tau,(p+1)/2,(p+1)/2-1}(q^{1/2}/\lambda),$$ (6.13)

and it follows that just exchanging the row $j$ with the row $p - j$, for any $j \in \{1,..,(p+1)/2\}$, and then the column $j$ with the column $p - j$, for any $j \in \{1,..,(p+1)/2\}$, in the matrix $\bar{D}_{\tau,(p+1)/2,(p+1)/2+1}(\lambda q^{1/2})$. Note that the determinants $det_{p-1} \bar{D}_{\tau,(p+1)/2,(p+1)/2+1}(\lambda q^{1/2})$ and $det_{p-1} \bar{D}_{\tau,(p+1)/2,(p+1)/2-1}(\lambda q^{1/2})$ are Laurent's rational functions both finite for $\lambda \to \pm 1$ and

$\lambda \to \pm i$. This implies that in both the limits $\lambda \to \pm 1$ and $\lambda \to \pm i$ the function $\det_p \bar{D}_\tau(\lambda q^{1/2})$ is finite. As $Z = 2$ and $Z = -2$ are the only points for which $\det_p \bar{D}_\tau(\lambda)$ may have divergencies, then, from $\tau(\lambda)$ and $\bar{x}(\lambda)$ Laurent's polynomial in $\lambda$ of degree $2N+4$, it follows that $\det_p \bar{D}_\tau(\lambda)$ is a polynomial of degree $N + 2$ in the variable $Z$. Let us now remark that by the SoV characterization of the spectrum we have that $\tau(\lambda) \in \Sigma_{\mathscr{T}}$ if and only if it has the form (5.1) and satisfies:

$$\det_p \bar{D}_\tau(\xi_a^{(0)}) = 0, \ \forall a \in \{1, ..., N\}, \tag{6.14}$$

as we are assuming that (6.7) is satisfied. Finally, it is simple to verify that the following asymptotic hold:

$$\det_p \left[ \lim_{\lambda \to +\infty} \lambda^{-2(N+2)} \bar{D}_\tau(\lambda) \right] = \det_p \left[ \lim_{\lambda \to 0} \lambda^{2(N+2)} \bar{D}_\tau(\lambda) \right]^t \tag{6.15}$$

$$= \det_p \begin{pmatrix} \tau_\infty & 0 & 0 & \cdots & 0 & -\bar{A}_\infty \\ -x\bar{A}_\infty & x\tau_\infty & 0 & 0 & \cdots & 0 \\ 0 & -x^2\bar{A}_\infty & x^2\tau_\infty & \ddots & & \vdots \\ \vdots & & \ddots & \ddots & 0 & 0 \\ 0 & \cdots & 0 & -x^{2l-1}\bar{A}_\infty & x^{2l-1}\tau_\infty & 0 \\ 0 & 0 & \cdots & 0 & -x^{2l}\bar{A}_\infty & x^{2l}\tau_\infty \end{pmatrix}, \tag{6.16}$$

where we have denoted with $^t$ the transpose of the matrix and $x = q^{2(N+2)}$, so that it holds:

$$\lim_{\log \lambda \to \pm\infty} \lambda^{\mp 2p(N+2)} \det_p \bar{D}_\tau(\lambda) = \tau_\infty^p - \bar{A}_\infty^p. \tag{6.17}$$

These results fix completely the Laurent's polynomial $\det_p \bar{D}_\tau(\lambda)$ to satisfy (6.9). $\quad\square$

Let us introduce now the following function:

$$G(\lambda|x, y) = F(\lambda) \left[ \left( \tau_\infty - q^{-2(p-1)N} \bar{A}_\infty \right) x \prod_{a=0}^{1} \left( \frac{\lambda}{i^a q^{1/2}} - \frac{i^a q^{1/2}}{\lambda} \right) \left( i^a q^{1/2} \lambda - \frac{1}{i^a q^{1/2} \lambda} \right) \right.$$
$$\left. + (i-1) \frac{y A(iq^{1/2})}{4F(iq^{1/2})} \prod_{a=0}^{1} \left( \lambda q^{(1-2a)/2} - \frac{1}{\lambda q^{(1-2a)/2}} \right) \right], \tag{6.18}$$

and the states:

$$\langle \omega | = \sum_{h_1,...,h_N=0}^{p-1} \prod_{a=1}^{N} \prod_{k_a=0}^{h_a-1} \frac{A(1/\zeta_a^{(k_a)})}{D(1/\zeta_a^{(k_a)})} \prod_{1 \le b < a \le N} (X_a^{(h_a)} - X_b^{(h_b)}) \langle h_1, ..., h_N |, \tag{6.19}$$

$$| \bar{\omega} \rangle = \sum_{h_1,...,h_N=0}^{p-1} \prod_{1 \le b < a \le N} (X_a^{(h_a)} - X_b^{(h_b)}) | h_1, ..., h_N \rangle, \tag{6.20}$$

and a renormalization of the operator

$$\hat{\mathscr{B}}_-(\lambda) = \frac{(\zeta_- - 1/\zeta_-)}{\kappa_- e^{\tau_-} (\lambda^2/q - q/\lambda^2) \prod_{n=1}^{N} \alpha_n \beta_n} \mathscr{B}_-(\lambda), \tag{6.21}$$

which is a degree $N$ polynomial in $\Lambda$. In the following we denote with $Q(\lambda)$ a polynomial in $\Lambda = \lambda^2 + 1/\lambda^2$ of degree $N_Q \le (p-1)N$ which admits the following interpolation formula:

$$Q(\lambda) = \sum_{a=1}^{(p-1)N+1} \prod_{b=1, b \ne a}^{(p-1)N+1} \frac{\Lambda - w_b}{w_a - w_b} Q(\xi_a), \tag{6.22}$$

where $w_b = \xi_b^2 + 1/\xi_b^2$ for any $b \in \{1, ..., (p-1)N+1\}$ and where we have defined:

$$\xi_{s(n,h_n)} \equiv \xi_n^{(h_n-1)}, \ \ s(n,h_n) \equiv (n-1)(p-1)+h_n, \ \ \forall n \in \{1,...,N\}, h_n \in \{1,...,p-1\}, \quad (6.23)$$

while $\xi_{(p-1)N+1}$ is arbitrary. Moreover, we use the notation:

$$q_\infty \equiv \sum_{a=1}^{(p-1)N+1} \prod_{b=1, b\neq a}^{(p-1)N+1} \frac{Q(\xi_a)}{w_a - w_b}, \ \ q_0 \equiv Q(iq^{1/2}). \quad (6.24)$$

Here, $q_\infty$ is the coefficient in $\Lambda^{(p-1)N}$ of the power expansion of the polynomial $Q(\lambda)$. Once this notation are introduced, the previous characterization of the spectrum can be reformulated in terms of Baxter's type TQ-functional equations and the eigenstates admit an algebraic Bethe ansatz like reformulation, as we show in the next theorem.

**Theorem 6.1.** *Let the conditions* (4.23)*,* (4.24)*,* (5.15) *and* (6.7) *be satisfied and let* $\tau(\lambda)$ *be an entire function for which there exists a polynomial* $Q(\lambda)$ *of the form* (6.22) *satisfying the conditions:*

$$Q(\xi_a^{(0)}) \neq 0, \ \ \forall a \in \{1,...,N\}, \quad (6.25)$$

*and the following functional equation:*

$$\tau(\lambda)Q(\lambda) = \overline{A}(\lambda)Q(\lambda/q) + \overline{A}(1/\lambda)Q(\lambda q) + G(\lambda|q_\infty, q_0), \quad (6.26)$$

*with* $(p-1)N-1 \leq N_Q$*, then* $\tau(\lambda) \in \Sigma_{\mathcal{T}}$ *and (up to normalization) the left and right transfer matrix eigenstates associated to it admit the following Bethe ansatz like representations:*

$$\langle \tau | = \langle \omega | \prod_{b=1}^{N_Q} \hat{\mathcal{B}}_-(\lambda_b), \ \ | \tau \rangle = \prod_{b=1}^{N_Q} \hat{\mathcal{B}}_-(\lambda_b) | \bar{\omega} \rangle, \quad (6.27)$$

*where the* $\lambda_b$ *(fixed up the symmetry* $\lambda_b \to -\lambda_b$*,* $\lambda_b \to 1/\lambda_b$*) for* $b \in \{1,...,N_Q\}$ *are the zeros of* $Q(\lambda)$*. Vice versa, if* $\tau(\lambda) \in \Sigma_{\mathcal{T}}$ *then there exists a polynomial* $Q(\lambda)$ *of the form* (6.22) *with* $(p-1)N-1 \leq N_Q$ *satisfying with* $\tau(\lambda)$ *the Baxter's equation* (6.26)*.*

*Proof.* The proof of this type of reformulation of the spectrum is by now quite standard and it has been proven for several models once they admit SoV description, see for example [36, 94–96, 101–103]. So we will try to point out just some main features of the proof. Let us start proving the first part of the statement. It is simple to remark that the r.h.s. of the equation (6.26) is a Laurent polynomial in $\lambda$, indeed we can write:

$$\overline{A}(\lambda)Q(\lambda/q) + \overline{A}(1/\lambda)Q(\lambda q) = \frac{\overline{x}(\lambda)Q(q/\lambda) - \overline{x}(1/\lambda)Q(\lambda q)}{\lambda^2 - 1/\lambda^2} \quad (6.28)$$

so that the limits $\lambda \to \pm 1$, $\lambda \to \pm i$ are all finite. Moreover, it is simple to observe that the r.h.s. of (6.26) is invariant under the transformations $\lambda \to -\lambda$ and $\lambda \to 1/\lambda$, so that the r.h.s. of (6.26) is in fact a polynomial of degree $N_Q + N + 2 \leq pN + 2$ in $\Lambda$. Then, the fact that $\tau(\lambda)$ is entire in $\lambda$ implies by the equation (6.26) that $\tau(\lambda)$ is a polynomial in $\Lambda$ of the form (5.1) with (5.14). This together with the equations:

$$\det_p \bar{D}_\tau(\xi_a^{(0)}) = 0, \ \ \forall a \in \{1,...,N\}, \quad (6.29)$$

which are trivial consequences of (6.26) and of (6.25), imply by the SoV characterization that $\tau(\lambda) \in \Sigma_{\mathcal{T}}$. Let us show now that the eigenstates associated to this $\tau(\lambda) \in \Sigma_{\mathcal{T}}$ can be written

in the form (6.27). In order to do so let us observe that the $Q(\lambda)$ which is solution of (6.26) is defined up to a constant factor so that we are free to fix it by writing:

$$Q(\lambda) = \prod_{b=1}^{N_Q} (\Lambda - \Lambda_b) \tag{6.30}$$

where $\Lambda_b = \lambda_b^2 + 1/\lambda_b^2$, then we have that it holds:

$$\prod_{b=1}^{N_Q} \hat{\mathscr{B}}_-(\lambda_b)|\bar{\omega}\rangle = \sum_{h_1,\dots,h_N=0}^{p-1} \prod_{b=1}^{N_Q}\prod_{a=1}^{N} (X_a^{(h_a)} - \Lambda_b) \prod_{1\leq b<a\leq N} (X_a^{(h_a)} - X_b^{(h_b)})|h_1,\dots,h_N\rangle$$

$$= \sum_{h_1,\dots,h_N=0}^{p-1} \prod_{a=1}^{N} Q(X_a^{(h_a)}) \prod_{1\leq b<a\leq N} (X_a^{(h_a)} - X_b^{(h_b)})|h_1,\dots,h_N\rangle, \tag{6.31}$$

where in the last line appears the SoV characterization of the right eigenstate associated to $\tau(\lambda) \in \Sigma_{\mathscr{T}}$. The same steps are used to prove (6.27) for the left eigenstate. Let us comment that the polynomiality of $Q(\lambda)$ is the central property that we have used to prove these rewriting of the SoV characterizations of the transfer matrix eigenstates. This type of rewriting was first observed in the case of the noncompact XXX chains [90]. It is quite general and it has been used already for different models in the SoV framework, see for example [101–103].

Let us now assume that $\tau(\lambda) \in \Sigma_{\mathscr{T}}$ and let us prove that it exists $Q(\lambda)$ of the form (6.22). In order to do so it is enough to show that the Baxter's equation holds in $pN + 2$ different values of $\Lambda$, that we chose to be $\Lambda = \pm(q + 1/q)$ and the $\Lambda = X_a^{(h_a)}$ for any $a \in \{1,\dots,N\}$ and $h_a \in \{0,\dots,p-1\}$. This set of equations can be organized in the following form:

$$\bar{D}_\tau(\xi_a^{(0)}) \begin{pmatrix} Q(\xi_a^{(h=0)}) \\ Q(\xi_a^{(h=1)}) \\ \vdots \\ \vdots \\ Q(\xi_a^{(h=p-1)}) \end{pmatrix}_{p\times 1} = \begin{pmatrix} 0 \\ \vdots \\ \vdots \\ \vdots \\ 0 \end{pmatrix}_{p\times 1} \qquad \forall a \in \{1,\dots,N\}. \tag{6.32}$$

They are equivalent to the following system of equations:

$$Q(\xi_a^{(0)})\tau(\xi_a^{(0)})/\overline{A}(1/\xi_a^{(0)}) = Q(\xi_a^{(1)}) \tag{6.33}$$

$$\frac{\tau(\xi_a^{(h)})}{\overline{A}(1/\xi_a^{(h)})}Q(\xi_a^{(h)}) - \frac{\overline{A}(\xi_a^{(h)})}{\overline{A}(1/\xi_a^{(h)})}Q(\xi_a^{(h-1)}) = Q(\xi_a^{(h+1)}), \ \forall h \in \{1,\dots,p-3\}, \tag{6.34}$$

$$\frac{\tau(\xi_a^{(p-2)})}{\overline{A}(1/\xi_a^{(p-2)})}Q(\xi_a^{(p-2)}) - \frac{\overline{A}(\xi_a^{(p-2)})}{\overline{A}(1/\xi_a^{(p-2)})}Q(\xi_a^{(p-3)}) = \prod_{b=1}^{(p-1)N} \frac{X_a^{(p-1)} - w_b}{w_{(p-1)N+1} - w_b}Q(\xi_{(p-1)N+1})$$

$$+ \sum_{n=1}^{N_Q} \prod_{\substack{b=1 \\ b\neq n}}^{(p-1)N+1} \frac{X_a^{(p-1)} - w_b}{w_n - w_b}Q(\xi_n), \quad \tag{6.35}$$

as $\tau(\lambda) \in \Sigma_{\mathscr{T}}$. So we are left with (6.33)-(6.35) a linear system of $(p-1)N$ inhomogeneous equations in the $(p-1)N$ unknowns $Q(\xi_a)$ for all $a \in \{1,\dots,(p-1)N\}$. Let us define by induction the following coefficients:

$$\mathbb{C}_{a,h+1} = x_{a,0,h}\mathbb{C}_{a,h} + x_{a,-,h}\mathbb{C}_{a,h-1}, \quad \forall h \in \{1,\dots,p-2\}, \ \mathbb{C}_{a,1} = \tau(\xi_a^{(0)})/\overline{A}(1/\xi_a^{(0)}), \ \mathbb{C}_{a,0} = 1, \tag{6.36}$$

where we have defined:

$$x_{a,0,h} = \frac{\tau(\xi_a^{(h)})}{\overline{A}(1/\xi_a^{(h)})}, \quad x_{a,-,h} = -\frac{\overline{A}(\xi_a^{(h)})}{\overline{A}(1/\xi_a^{(h)})}, \quad \forall h \in \{1,...,p-2\}, \ \forall a \in \{1,...,N\}. \tag{6.37}$$

One can rewrite the previous system as it follows:

$$Q(\xi_a^{(h)}) = \mathbb{C}_{a,h} Q(\xi_a^{(0)}), \quad \forall h \in \{1,...,p-1\}, \ \forall a \in \{1,...,N\}, \tag{6.38}$$

and

$$\mathbb{C}_{a,p-1} Q(\xi_a^{(0)}) = \prod_{b=1}^{(p-1)N} \frac{X_a^{(p-1)} - w_b}{w_{(p-1)N+1} - w_b} Q(\xi_{(p-1)N+1}) + \sum_{n=1}^{N} \mathbb{D}_{a,n} Q(\xi_n^{(0)}), \tag{6.39}$$

where

$$\mathbb{D}_{b,n} = \sum_{h=0}^{p-2} \prod_{c=1, c \neq a(n,h)}^{(p-1)N+1} \frac{X_b^{(p-1)} - w_c}{w_n - w_c} \mathbb{C}_{n,h}. \tag{6.40}$$

So finally we are left with a system of N inhomogeneous equations in the N unknown $Q(\xi_a^{(0)})$ for all $a \in \{1,...,N\}$, which always admits a nontrivial solution, i.e.

$$\{Q(\xi_1^{(0)}),...,Q(\xi_N^{(0)})\} \neq \{0,...,0\}. \tag{6.41}$$

Finally, let us point out that the degree of the polynomial $Q(\lambda)$ is constrained by the Baxter's equation (6.26). Indeed, it is trivial to remark that if $N_Q \leq (p-1)N-2$, then the equation (6.26) admits only the trivial solution $Q(\lambda) = 0$ which is not compatible with equation (6.41). Instead, the equation (6.26) may still be satisfied with a nontrivial $Q(\lambda)$ if $N_Q = (p-1)N-1$ but only if the following condition:

$$\lim_{\Lambda \to \infty} \Lambda^{(p-1)N-1} Q(\lambda) = \frac{(i-1)A(iq^{1/2})q_0}{4\left(\tau_\infty - q^{2-2(p-1)N}\overline{A}_\infty\right)F(iq^{1/2})}, \tag{6.42}$$

is satisfied. It is clear that this represents one supplementary condition and one can expect that up to some exceptional case, related to the values of the parameters of the representation and to some special choice of the $\tau(\lambda) \in \Sigma_{\mathcal{T}}$, it is not satisfied and so that $N_Q = (p-1)N$. $\quad\square$

It is then simple to prove the following corollary which provides under some further constraints a complete characterization of the transfer matrix eigenvalues and eigenstates in terms of solution of ordinary Bethe equations.

**Corollary 6.1.** *If the conditions (4.23), (4.24), (5.15) and (6.7) are satisfied and if we fix the boundary parameter $\zeta_+ = iz_+$ with $z_+ \in \{-1,+1\}$ and the following global condition:*

$$\frac{\kappa_+ e^{\epsilon(\tau_- - \tau_+)}}{iz_+ \alpha_- \beta_-} = q^{1+N} \prod_{n=1}^{N} \frac{b_n c_n}{\alpha_n \beta_n}, \tag{6.43}$$

*where $\epsilon = +1$ for $b_+(\lambda) = 0$, $c_+(\lambda) \neq 0$ and $\epsilon = -1$ for $c_+(\lambda) = 0$, $b_+(\lambda) \neq 0$, then $\mathcal{T}(\lambda)$ has simple spectrum and $\tau(\lambda) \in \Sigma_{\mathcal{T}}$ if and only if $\tau(\lambda)$ is entire and there exists a polynomial $Q(\lambda)$ of the form (6.22), with $N_Q \leq (p-1)N$, which satisfies the following homogeneous Baxter equation:*

$$\tau(\lambda)Q(\lambda) = \overline{A}(\lambda)Q(\lambda/q) + \overline{A}(1/\lambda)Q(\lambda q). \tag{6.44}$$

*The $(\lambda_1,...,\lambda_{N_Q})$ entering in the Bethe ansatz like representations of the eigenstates (6.27) are solutions of the associated ordinary Bethe equations.*

*Proof.* The condition $\zeta_+ = iz_+$ with $z_+ \in \{-1, +1\}$ implies:

$$\tau(iq^{1/2}) = A(iq^{1/2}) = 0, \tag{6.45}$$

so that the condition (6.43) implies that $G(\lambda|q_\infty, q_1) = 0$, for any choice of $q_\infty$, $q_1$. Then, the previous theorem directly implies our corollary. Finally, let us remark that in this case we do not have any restriction on the degree of the polynomial $Q(\lambda)$ imposed by the Baxter's equation (6.26), so that we are only left with $N_Q \leq (p-1)N$. □

## 7  Conclusions

In this paper we have studied the transfer matrix spectrum of the class of cyclic 6-vertex representations of the reflection algebra in the case of one completely general and one triangular boundary matrices and for bulk parameters satisfying some specific constraints. Our result is the complete characterization of the spectrum (eigenvalues and eigenstates) of this class of models both by a discrete system of Baxter's like second order difference equations and by a single inhomogeneous TQ-functional equation within a class of polynomial Q-functions.

The present paper represents a natural starting point to solve the spectral problem in the most general setting. In order to do so we need to generalize to the 6-vertex cyclic representations the Baxter's gauge transformations used in the 6-vertex spin 1/2 highest weight representations and prove then the pseudo-diagonalizability of the associated family of gauge transformed $\mathscr{B}_-$-operators. These points are currently under analysis.

An interesting point to which we would like to come back in future investigations is the explicit construction of the Q-operator for the cyclic representations of the 6-vertex reflection algebra. This can lead to new connections with transfer matrices of exactly solvable class of models of non 6-vertex type. Indeed, let us recall that for the special class of 6-vertex cyclic representations of the Yang-Baxter algebra, parametrized by points on the algebraic curve (A.37), the associated integrable models have some remarkable connections with the inhomogeneous p-state chiral Potts models. Indeed, the chiral Potts transfer matrices play the role of the Q-operators for the transfer matrix associated to the cyclic representations of the 6-vertex Yang-Baxter algebra [113].

## Acknowledgements

J. M. M. and G. N. are supported by CNRS and ENS Lyon; B. P. is supported by ENS Lyon and ENS Cachan.

## A  Appendix

### A.1  General proof of diagonalizability of $\mathscr{B}_-(\lambda)$

The generator $\mathscr{B}_-(\lambda)$ admits the following boundary bulk decomposition:

$$\mathscr{B}_-(\lambda) = -a_-(\lambda)A(\lambda)B(1/\lambda) + b_-(\lambda)A(\lambda)A(1/\lambda) - c_-(\lambda)B(\lambda)B(1/\lambda) + d_-(\lambda)B(\lambda)A(1/\lambda), \tag{A.1}$$

if the inner boundary matrix is non-diagonal and the bulk parameters are generals, i.e. if it holds:

$$B_- = (-1)^N \kappa_- e^{\tau_-} \prod_{a=1}^{N} \alpha_n \beta_n \neq 0, \tag{A.2}$$

then it trivially follows that $\mathscr{B}_-(\lambda)$ admits the following functional form:

$$\mathscr{B}_-(\lambda) = \mathrm{B}_-(\frac{\lambda^2}{q} - \frac{q}{\lambda^2}) \prod_{a=1}^{\mathsf{N}} (\frac{\lambda}{\mathscr{B}_{-,a}} - \frac{\mathscr{B}_{-,a}}{\lambda})(\lambda \mathscr{B}_{-,a} - \frac{1}{\lambda \mathscr{B}_{-,a}}), \tag{A.3}$$

where the $\mathscr{B}_{-,a}$ are invertible commuting operators. The above functional form implies that under the condition (A.2) the operator family is not nilpotent, in particular does not exist any state annihilated by $\mathscr{B}_-(\lambda)$ for any $\lambda \in \mathbb{C}$. That is does not exist a reference state and we cannot use ABA to analyze the spectral problem of the transfer matrix. Here, we show that under the condition (A.2) for general values of the bulk parameters the $\mathscr{B}_-(\lambda)$ is indeed diagonalizable and with simple spectrum. Let us first prove the following lemma:

**Lemma A.1.** *There exists at least one simultaneous eigenstate:*

$$|\Omega_R\rangle, \quad \langle\Omega_L| \tag{A.4}$$

*of the one parameter family of commuting operators $\mathscr{B}_-(\lambda)$:*

$$\mathscr{B}_-(\lambda)|\Omega_R\rangle = |\Omega_R\rangle \mathrm{B}_-(\frac{\lambda^2}{q} - \frac{q}{\lambda^2}) \prod_{a=1}^{\mathsf{N}} (\frac{\lambda}{\bar{b}_{-,a}} - \frac{\bar{b}_{-,a}}{\lambda})(\lambda \bar{b}_{-,a} - \frac{1}{\lambda \bar{b}_{-,a}}) \tag{A.5}$$

$$\langle\Omega_L|\mathscr{B}_-(\lambda) = \mathrm{B}_-(\frac{\lambda^2}{q} - \frac{q}{\lambda^2}) \prod_{a=1}^{\mathsf{N}} (\frac{\lambda}{\hat{b}_{-,a}} - \frac{\hat{b}_{-,a}}{\lambda})(\lambda \hat{b}_{-,a} - \frac{1}{\lambda \hat{b}_{-,a}})\langle\Omega_L|. \tag{A.6}$$

*Proof.* We can always put $\mathscr{B}_{-,1}$ in a Jordan normal form, let us denote with $\mathrm{B}_1$ the right eigenspace associated to a given eigenvalue $\bar{b}_{-,1} \neq 0$ of $\mathscr{B}_{-,1}$, as $\mathscr{B}_{-,1}$ is invertible. If this eigenspace is one dimensional we have found our simultaneous eigenstate of $\mathscr{B}_-(\lambda)$. If this is not the case then by the commutativity we have that $\mathrm{B}_1$ is an invariant space with regards to $\mathscr{B}_{-,n}$ for any $n \in \{1, ..., N\}$. So we can always put $\mathscr{B}_{-,2}$ in a Jordan normal form in $\mathrm{B}_1$, let us denote with $\mathrm{B}_{1,2}$ the eigenspace associated to a given eigenvalue $\bar{b}_{-,2} \neq 0$ of $\mathscr{B}_{-,2}$. If this eigenspace is one dimensional we have found our simultaneous eigenstate of $\mathscr{B}_-(\lambda)$ otherwise we can reiterate this procedure. We can have two possibilities both at the step $n \leq N$ we find that the eigenspace $\mathrm{B}_{1,...,n}$ is one-dimensional or we arrive up to the eigenspace $\mathrm{B}_{1,...,N}$ in both the cases we have (at least) one simultaneous eigenstate of the $\mathscr{B}_{-,n}$ for any $n \in \{1, ..., N\}$ and so one eigenstate of $\mathscr{B}_-(\lambda)$. Similarly, we can prove the existence of $\langle\Omega_L|$. $\square$

From the previous Lemma and the reflection algebra equation we can prove the following proposition.

**Proposition A.1.** *Under the condition (A.2) for almost all the values of the bulk parameters, the operator family $\mathscr{B}_-(\lambda)$ is diagonalizable and it has simple spectrum and its average value is central and it holds:*

$$\mathbb{B}_-(\lambda) = \prod_{a=1}^{p} \mathscr{B}_-(\lambda q^a) = \mathrm{B}_-^p(\frac{\lambda^{2p}}{1} - \frac{1}{\lambda^{2p}}) \prod_{a=1}^{\mathsf{N}} (\frac{\lambda^p}{\bar{b}_{-,a}^p} - \frac{\bar{b}_{-,a}^p}{\lambda^p})(\lambda^p \bar{b}_{-,a}^p - \frac{1}{\lambda^p \bar{b}_{-,a}^p}), \tag{A.7}$$

*with:*

$$\bar{b}_{-,m}^p \neq \bar{b}_{-,n}^p, \ \forall n \neq m \in \{1, ..., N\}. \tag{A.8}$$

*Proof.* Let us observe that by definition the operator family $\mathscr{B}_-(\lambda)$ is a polynomial in the bulk parameters so the same must be true for its spectrum. This implies in particular that defined

$$\hat{b}_{n,m} = \bar{b}_{-,n}^p - \bar{b}_{-,m}^p, \ \forall n \neq m \in \{1, ..., N\}, \tag{A.9}$$

or they are identically zero or they can be zero only over subspaces of nonzero codimension in the space of the bulk parameters. So here we have just to prove that the $\hat{b}_{n,m}$ are not identically zero to derive that (A.8) holds for almost all the values of the parameters. To do so we can just recall that from the results derived in the previous section it holds:

$$\bar{b}^p_{-,m} = q^{p/2}\mu^p_{m,+}, \ \forall m \in \{1,...,N\},\tag{A.10}$$

under the condition $B(\lambda)$ nilpotent, i.e. $a^p_n = -b^p_n$ for any $n \in \{1,...,N\}$. From which the condition (A.8) holds as soon as we impose:

$$\beta^p_n/\alpha^p_n \neq \beta^p_m/\alpha^p_m, \ \forall n \neq m \in \{1,...,N\}.\tag{A.11}$$

Once we have proven this statement, then we have just to use the reflection algebra to construct an eigenbasis of $\mathscr{B}_-(\lambda)$ and this is done just by a repeated action of the generators $\mathscr{A}_-(\lambda)$ computed in the zeros of $\mathscr{B}_-(\lambda)$ on the eigenstates $|\Omega_R\rangle$ and $\langle\Omega_L|$. That is we repeat the construction of the eigenbasis presented in Section 4.2 by substituting to the reference states defined in (4.1) with the $\mathscr{B}_-(\lambda)$ on the eigenstates $|\Omega_R\rangle$ the state $\langle\Omega_L|$. Note that such an action can generate a null vector only if some of the zeros of $\mathscr{B}_-(\lambda)$ coincides with the zeros of the quantum determinant anyhow as discussed in Section 4.2 under the condition (4.24) we are always able to chose an appropriate set of $p^N$ zeros of $\mathscr{B}_-(\lambda)$ which do not coincides with those of the quantum determinant so that we generate exactly $p^N$ independent states. $\qquad\square$

## A.2 The lattice sine-Gordon model with integrable boundaries

In this appendix we point out that the results on the transfer matrix spectrum for general cyclic representations developed in this paper apply also to characterize the spectrum of the transfer matrix of the lattice sine-Gordon model with integrable open boundary conditions. Let us recall that the Lax operator defining the lattice sine-Gordon model has the following form:

$$L^{sG}_{a,n}(\lambda|\mathrm{U}_n,\mathrm{V}_n,\kappa_n,r_n,s_n) \equiv \begin{pmatrix} \mathrm{U}_n\left(q^{-1/2}\kappa^2_n r_n s_n \mathrm{V}_n + \frac{q^{1/2}s_n}{\mathrm{V}_n r_n}\right) & \frac{\kappa_n}{i}\left(\lambda r_n \mathrm{V}_n - \frac{1}{\lambda r_n \mathrm{V}_n}\right) \\ \frac{\kappa_n}{i}\left(\frac{\lambda}{r_n \mathrm{V}_n} - \frac{r_n \mathrm{V}_n}{\lambda}\right) & \mathrm{U}^{-1}_n\left(\frac{q^{1/2}r_n \mathrm{V}_n}{s_n} + \frac{q^{-1/2}\kappa^2_n}{s_n r_n \mathrm{V}_n}\right) \end{pmatrix}_a,\tag{A.12}$$

each local representation being defined as the representation of a local Weyl algebra

$$\mathrm{U}_n\mathrm{V}_m = q^{\delta_{n,m}}\mathrm{V}_m\mathrm{U}_n \quad \forall n,m \in \{1,...,N\},\tag{A.13}$$

associated to a root of unit $q$, where $\mathrm{U}_n$ and $\mathrm{V}_n$ are the Weyl algebra generators on the Hilbert space $\mathscr{R}_n$. Let us introduce the monodromy matrices for the Yang-Baxter and reflection equations of the lattice sine-Gordon model, they read:

$$\begin{aligned} M^{sG}_a(\lambda) &= L_{a,N}(\lambda q^{-1/2}/\xi_N)\cdots L_{a,1}(\lambda q^{-1/2}/\xi_1) \in \mathrm{End}(\mathbb{C}^2 \otimes \mathscr{H}), &(A.14)\\ \mathscr{U}^{sG}_{a,-}(\lambda) &= M^{sG}_a(\lambda)K_{a,-}(\lambda)\hat{M}^{sG}_a(\lambda) \in \mathrm{End}(\mathbb{C}^2 \otimes \mathscr{H}), &(A.15) \end{aligned}$$

and also the boundary transfer matrix of the sine-Gordon model, which reads:

$$\mathscr{T}^{sG}(\lambda) \equiv tr_a\{K_{a,+}(\lambda)\mathscr{U}^{sG}_{a,-}(\lambda)\}.\tag{A.16}$$

Let us remark that we have used the upper index sG in the above two monodromy matrices and transfer matrix to point out that they are related to the sine-Gordon model while we will continue to denote with $M_a(\lambda)$, $\mathscr{U}_{a,-}(\lambda)$ and $\mathscr{T}(\lambda)$ those associated to the original $\tau_2$-model. The following lemma establishes the connection between the monodromy and transfer matrices of the sine-Gordon model and the original $\tau_2$-model.

**Lemma A.2.** *Let us denote* $\mathsf{N} = 2\mathsf{M} + \mathsf{x}$ *with* $\mathsf{x} \in \{0, 1\}$ *and let us impose the following identification of the generators of the local Weyl algebras:*

$$\mathsf{U}_{2n+\mathsf{y}} = u_{2n+\mathsf{y}}^{(1-2\mathsf{x})(1-2\mathsf{y})}, \quad \mathsf{V}_{2n+\mathsf{y}} = v_{2n+\mathsf{y}}^{(1-2\mathsf{x})(1-2\mathsf{y})}, \tag{A.17}$$

*with* $\mathsf{y} \in \{0, 1\}$ *and* $2n + \mathsf{y} \in \{1, ..., \mathsf{N}\}$. *Then the following identity holds:*

$$M_a^{sG}(\lambda) = M_a(\lambda)\left(\sigma_a^x\right)^{\mathsf{x}}, \tag{A.18}$$

*once we define the parameters of the* $\tau_2$*-model in terms of those of the lattice sine-Gordon model as it follows:*

$$a_{2n+\mathsf{y}} = \kappa_{2n+\mathsf{y}}^2 \left(r_{2n+\mathsf{y}}s_{2n+\mathsf{y}}\right)^{(1-2\mathsf{x})(1-2\mathsf{y})}, \qquad b_{2n+\mathsf{y}} = \left(\frac{s_{2n+\mathsf{y}}}{r_{2n+\mathsf{y}}}\right)^{(1-2\mathsf{x})(1-2\mathsf{y})}, \tag{A.19}$$

$$c_{2n+\mathsf{y}} = \left(\frac{r_{2n+\mathsf{y}}}{s_{2n+\mathsf{y}}}\right)^{(1-2\mathsf{x})(1-2\mathsf{y})}, \qquad d_{2n+\mathsf{y}} = \frac{\kappa_{2n+\mathsf{y}}^2}{\left(r_{2n+\mathsf{y}}s_{2n+\mathsf{y}}\right)^{(1-2\mathsf{x})(1-2\mathsf{y})}}, \tag{A.20}$$

$$\alpha_{2n+\mathsf{y}} = \frac{\kappa_{2n+\mathsf{y}} r_{2n+\mathsf{y}}^{(1-2\mathsf{x})(1-2\mathsf{y})}}{i\xi_{2n+\mathsf{y}}}, \qquad \beta_{2n+\mathsf{y}} = \frac{\kappa_{2n+\mathsf{y}}\xi_{2n+\mathsf{y}}}{ir_{2n+\mathsf{y}}^{(1-2\mathsf{x})(1-2\mathsf{y})}}. \tag{A.21}$$

*Moreover, under the above conditions and imposing the following identifications:*

$$\tau_\epsilon = \epsilon^{\mathsf{x}}\tau_\epsilon^{sG}, \quad \kappa_\epsilon = \epsilon^{\mathsf{x}}\kappa_\epsilon^{sG}, \quad \zeta_\epsilon = \left(\zeta_\epsilon^{sG}\right)^{\epsilon^{\mathsf{x}}} \text{ for } \epsilon = \pm, \tag{A.22}$$

*of the boundary parameters of the* $\tau_2$*-model and the lattice sine-Gordon model, we get:*

$$\mathscr{U}_{a,-}(\lambda) = \mathscr{U}_{a,-}^{sG}(\lambda), \quad \mathscr{T}(\lambda) = \mathscr{T}^{sG}(\lambda). \tag{A.23}$$

*Proof.* It is simple to observe that the following identities hold:

$$L_{a,n}^{sG}(\lambda/\xi_n | \mathsf{U}_n, \mathsf{V}_n, \kappa_n, r_n, s_n) = L_{a,n}(\lambda)\sigma_a^x \tag{A.24}$$

if:

$$\mathsf{U}_n = u_n, \quad \mathsf{V}_n = v_n, \tag{A.25}$$

and

$$a_n = \kappa_n^2 r_n s_n, \quad b_n = s_n/r_n, \quad c_n = r_n/s_n, \tag{A.26}$$

$$d_n = \kappa_n^2/(r_n s_n), \quad \alpha_n = -i\kappa_n r_n/\xi_n, \quad \beta_n = -i\kappa_n \xi_n/r_n. \tag{A.27}$$

Similarly by direct computations it is easy to show that defined:

$$\tilde{L}_{a,n}^{sG}(\lambda) \equiv \sigma_a^x L_{a,n}^{sG}(\lambda | \mathsf{U}_n, \mathsf{V}_n, \kappa_n, r_n, s_n)\sigma_a^x \tag{A.28}$$

it holds:

$$\tilde{L}_{a,n}^{sG}(\lambda) = L_{a,n}^{sG}(\lambda | \mathsf{U}_n^{-1}, \mathsf{V}_n^{-1}, \kappa_n, r_n^{-1}, s_n^{-1}). \tag{A.29}$$

Let us now observe that for $\mathsf{x} = 0$, we can write:

$$M_a^{sG}(\lambda) = [L_{a,2\mathsf{M}}(\frac{\lambda q^{-1/2}}{\xi_{2\mathsf{M}}})\sigma_a^x][\tilde{L}_{a,2\mathsf{M}-1}(\frac{\lambda q^{-1/2}}{\xi_{2\mathsf{M}-1}})\sigma_a^x]\cdots[L_{a,2}(\frac{\lambda q^{-1/2}}{\xi_2})\sigma_a^x][\tilde{L}_{a,1}(\frac{\lambda q^{-1/2}}{\xi_1})\sigma_a^x] \tag{A.30}$$

while for $x = 1$, we can write:

$$M_a^{sG}(\lambda)\sigma_a^x = [L_{a,2M+1}(\frac{\lambda q^{-1/2}}{\xi_{2M+1}})\sigma_a^x][\tilde{L}_{a,2M}(\frac{\lambda q^{-1/2}}{\xi_{2M}})\sigma_a^x]\cdots[\tilde{L}_{a,2}(\frac{\lambda q^{-1/2}}{\xi_2})\sigma_a^x][L_{a,1}(\frac{\lambda q^{-1/2}}{\xi_1})\sigma_a^x],$$

(A.31)

then these two identities together with the identity (*A.24*), (*A.29*) and the parametrization (*A.26*)-(*A.27*) imply the identity (*A.18*) with the parametrization (*A.19*)-(*A.21*)and the local operator identification (*A.17*). Finally, let us observe that from the identity (*A.18*) it follows:

$$\hat{M}_a^{sG}(\lambda) \equiv (-1)^N \sigma_a^y \left(M_a^{sG}(1/\lambda)\right)^{t_0} \sigma_a^y = \left(-\sigma_a^x\right)^x \hat{M}_a(\lambda),$$

(A.32)

which together with:

$$K_{a,+}(\lambda|\tau_\epsilon,\ \kappa_\epsilon,\ \zeta_\epsilon) = \left(\sigma_a^x\right)^x K_{a,+}^{sG}(\lambda|\tau_-^{sG},\kappa_-^{sG},\zeta_-^{sG})\left(-\sigma_a^x\right)^x,$$

(A.33)

holding under the parametrization (*A.22*), implies the identities (*A.23*). □

## A.3 Reduction to inhomogeneous chiral Potts representations

In this appendix we want to point out that a nontrivial class of representations of the inhomogeneous chiral Potts model can be described on the closed chain in the space of the parameters of the $\tau_2$-model considered in this paper. In order to do so let us recall that the transfer matrix $T_\lambda^{chP}$ of the inhomogeneous chiral Potts model [113] is characterized by the following kernel:

$$T_\lambda^{chP}(z,z') \equiv \langle z|T_\lambda^{chP}|z'\rangle = \prod_{n=1}^N W_{q_n p}(z_n/z_n')\bar{W}_{r_n p}(z_n/z_{n+1}'),$$

(A.34)

in the basis $\langle z| \equiv \langle z_1,...,z_N|$ and $|z\rangle \equiv |z_1,...,z_N\rangle$ formed by the left and right $u_n$-eigenstates:

$$\langle z|u_n = z_n\langle z|,\quad u_n|z\rangle = z_n|z\rangle \text{ with } z_n \in \mathbb{S}_p \equiv \{q^{2r},\ r = 1,..,p\},$$

(A.35)

and where:

$$\lambda = t_p^{-1/2}c_0,\quad p,\ r_n,\ q_n \in \mathscr{C}_k, c_0 \in \mathbb{C}.$$

(A.36)

The algebraic curve $\mathscr{C}_k$ of modulus $k$ is by definition the locus of the points in the four-dimensional complex space $p \equiv (a_p, b_p, c_p, d_p) \in \mathbb{C}^4$ which satisfy the equations:

$$x_p^p + y_p^p = k(1 + x_p^p y_p^p),\quad kx_p^p = 1 - k's_p^{-p},\quad ky_p^p = 1 - k's_p^p,$$

(A.37)

where:

$$x_p \equiv a_p/d_p,\quad y_p \equiv b_p/c_p,\quad s_p \equiv d_p/c_p, t_p \equiv x_p y_p,\quad k^2 + (k')^2 = 1,$$

(A.38)

and $W_{qp}(z(n))$ and $\bar{W}_{qp}(z(n))$ are the Boltzmann weights of the chiral Potts model. Then, $T_\lambda^{chP}$ is a Baxter Q-operator with regards to the bulk $\tau_2$-transfer matrix in $\mathscr{H}_N$.

Let us here directly characterize the class of the inhomogeneous chiral Potts representations once we restrict the space of the parameters to that used in the section 6; in particular, we assume that it holds:

$$b_n = -q^{-1}a_n,\ d_n = -q^{-1}c_n.$$

(A.39)

The parameters of the $\tau_2$-Lax operators are written in terms of the coordinate of the points p, $r_n$, $q_n$ by using the equations (5.3) of the paper [96]. Then we have that the points $r_n$, $q_n$ are elements of $\mathscr{C}_k$ if and only if beyond (*A.39*) the parameters of the $\tau_2$-Lax operators satisfy the following conditions:

$$\alpha_n\beta_n = a_n c_n$$

(A.40)

and

$$\left(\frac{c_0\alpha_n}{q^{1/2}a_n}\right)^p + \left(\frac{q^{1/2}c_0\alpha_n}{c_n}\right)^p = k\left(1 + \left(\frac{c_0^2\alpha_n^2}{c_n a_n}\right)^p\right). \tag{A.41}$$

Under these constraints the class of the inhomogeneous chiral Potts representations is characterized by the following identity:

$$r_n = \Delta(q_n), \quad \forall n \in \{1, ..., N\}, \tag{A.42}$$

where the $q_n$ are free elements of $\mathscr{C}_k$ and $\Delta$ is the following discrete automorphism of the curve:

$$\Delta : x = (a_x, b_x, c_x, d_x) \in \mathscr{C}_k \rightarrow \Delta(x) = (b_x, a_x, d_x, c_x) \in \mathscr{C}_k, \tag{A.43}$$

which implies:

$$x_{p_n} = y_{q_n}, \ y_{p_n} = x_{q_n}, \ s_{p_n} = s_{q_n}^{-1}. \tag{A.44}$$

Finally, this class of representations reduce to the superintegrable chiral Potts model under the following special homogeneous limits:

$$x_{q_n}^p \rightarrow (1 + k')/k, \quad \forall n \in \{1, ..., N\}. \tag{A.45}$$

## A.4 Reduction to the XXZ spin $s$ open chains at the $p = 2s + 1$ roots of unit

Here we show that imposing a set of conditions on the parameters of the $\tau_2$-Lax operator we can reduce it to the one of the spin $s = (p-1)/2$ XXZ case at the $p$ roots of unit. This has the interesting consequence that the analysis done of the open $\tau_2$-chain reduces for these special representations to that of an open spin chain under the same boundary conditions. In particular, we have that our functional equation characterization of the spectrum under a special homogeneous limit defines the spectrum of the following local Hamiltonian given by fusion and the Sklyanin formula:

$$\mathscr{H}_s = c_0 \frac{d}{d\lambda} \mathscr{T}_p(\lambda)\Big|_{\lambda=q^s} + \text{constant}, \tag{A.46}$$

with

$$\mathscr{H}_s = \sum_{n=1}^{N-1} H_{n,n+1}^{(s)} + \frac{d}{d\lambda} K_{1,-}^{(p)}(q^s) + \frac{\text{tr}_0\{K_{0,+}^{(p)}(q^s)H_{0,N}^{(s)}\}}{\text{tr}_0\{K_{0,+}^{(p)}(q^s)\}}, \tag{A.47}$$

where $\mathscr{T}_p(\lambda)$ is the $p$-fused open transfer matrix, $H_{n,n+1}^{(s)}$ is the two sites local Hamiltonian of the spin $s$ XXZ chain, $K_{0,\pm}^{(p)}(\lambda)$ are the $p \times p$ matrices $p$-fused scalar solutions of the reflection algebra obtained by doing the fusion $p-1$ times starting from the original $2 \times 2$ scalar solutions $K_{0,\pm}(\lambda)$, respectively.

**Lemma A.3.** *Let us fix the parameters of the $\tau_2$-representations as follows:*

$$\alpha_n = \beta_n = 1/2, \ a_n = q^{-1/2}/2i, \tag{A.48}$$

$$b_n = iq^{1/2}/2, \ c_n = q^{-1/2}/2i, \ d_n = iq^{1/2}/2, \tag{A.49}$$

*and let*

$$L_{a,n}^{XXZ}(\lambda) = \begin{pmatrix} \left[\lambda q^{s+S_n^z/2} - 1/(\lambda q^{s+S_n^z/2})\right]/2 & S_n^- \\ S_n^+ & \left[\lambda q^{s-S_n^z/2} - 1/(\lambda q^{s-S_n^z/2})\right]/2 \end{pmatrix} \tag{A.50}$$

*be the Lax operator of the spin s XXZ chain with anisotropy $\cosh\eta = (q + 1/q)/2$, then it holds:*

$$L_{a,n}^{XXZ}(\lambda) = L_{a,n}(\lambda/q) \tag{A.51}$$

for $s = (p-1)/2$ and $q = e^{i\pi p'/p}$ with $p'$ even and $p$ odd and coprime, which is equivalent to the following identities among the generators of the local algebras:

$$S_n^+ = u_n^{-1}(v_n - 1/v_n)/2i, \quad S_n^- = u_n(v_n/q - q/v_n)/2i, \tag{A.52}$$

and

$$S_n^z = \frac{2p}{i\pi p'}\log v_n - (p+1) \in \{-2s, -2(s-1), ..., 2s\} \mod 2p \tag{A.53}$$

*Proof.* Let us denote

$$\overline{|a,n\rangle} = \begin{pmatrix} 0 & \cdots & 1 & \cdots & 0 \end{pmatrix}^{t_0}, \quad a \in \{1, ..., 2s+1\} \tag{A.54}$$

the element $a$ of the canonical basis given by the column vector with all elements zero except that in row $a$ which is 1. In this basis we have the following representations for the generators

$$S_n^+ = \begin{pmatrix} 0 & f(1) & & \\ & \ddots & & \\ & & \ddots & f(2s) \\ & & & 0 \end{pmatrix}, \quad S_n^- = \begin{pmatrix} 0 & & & \\ f(1) & \ddots & & \\ & \ddots & \ddots & \\ & & f(2s) & 0 \end{pmatrix}, \tag{A.55}$$

where:

$$f(j) = \sqrt{\sinh j\eta \sinh(p-j)\eta} = i(q^j - q^{-j})/2, \tag{A.56}$$

and the second identity holds for $q^p = 1$, and:

$$S_n^z = \begin{pmatrix} 2s & 0 & & \\ 0 & \ddots & \ddots & \\ & \ddots & \ddots & 0 \\ & & 0 & -2s \end{pmatrix}. \tag{A.57}$$

Let us now impose that in our representation the $v_n$-eigenstates coincide with the elements of the canonical basis:

$$|p-a, n\rangle = \overline{|a+1, n\rangle} \quad \forall a \in \{0, ..., p-1\}, \tag{A.58}$$

we can verify now the formulae (A.52)-(A.53). The formula (A.53) is equivalent to:

$$v_n = q^{(S_n^z + p + 1)/2} \tag{A.59}$$

which holds for the following identities:

$$q^{(S_n^z + p + 1)/2}\overline{|a+1, n\rangle} = \overline{|a+1, n\rangle}q^{(2(s-a)+p+1)/2} = |p-a, n\rangle q^{p-a} = v_n|p-a, n\rangle. \tag{A.60}$$

Similarly we have:

$$S_n^+\overline{|a, n\rangle} = \overline{|a+1, n\rangle}f(a) = \overline{|a+1, n\rangle}(q^{-a} - q^a)/2i \tag{A.61}$$

$$= \left[u_n^{-1}(v_n - 1/v_n)/2i\right]|p-a, n\rangle \quad \forall a \in \{1, ..., p\} \tag{A.62}$$

and

$$S_n^-\overline{|a, n\rangle} = \overline{|a-1, n\rangle}f(a-1) = \overline{|a-1, n\rangle}(q^{1-a} - q^{a-1})/2i \tag{A.63}$$

$$= \left[u_n(v_n/q - q/v_n)/2i\right]|p+2-a, n\rangle \quad \forall a \in \{1, ..., p\}. \tag{A.64}$$

$\square$

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
