# Peer review of "Transfer matrix spectrum for cyclic representations of the 6-vertex reflection algebra I"

_SciPost Physics, doi:SciPost Phys. 2, 009 (2017)_

## Round 2 · Referee Report · Anonymous (Referee 2) · 2017-1-13

Strengths

See the report

Weaknesses

See the report

Report

The authors study the spectrum of Sklyanin's transfer matrix in the case of the cyclic representation, in the context of the 6-vertex model. This problem is also related to the spectrum of important statistical models (at q root of unity) as well as the lattice sine-Gordon model with open boundary conditions.

More precisely, they consider the case of generic non-diagonal boundary conditions and they determine the eigenvalues and eigenfunctions of the open transfer matrix by means of the separations of variables method. The connection of the results obtained via the SoV with Baxter's TQ functional relations is also provided and is one of the important results of this article.

This is an interesting work and may be seen as a continuation/generalization of previous works on the subject and some earlier works of some of the authors e.g.: [21], [37], [32], [35]....

Perhaps the authors should also note: A. Doikou,J.Stat.Mech.0609:P05010,2006,
where relevant findings regarding the XXZ model at q root of unity as well as the lattice sine-Gordon model are presented using generic local gauge transformations.

In summary, this article contains significant new results, with applications to a variety of models, therefore I propose its publication to SciPost.

Requested changes

No changes are requested.

  • validity: high
  • significance: high
  • originality: good
  • clarity: top
  • formatting: excellent
  • grammar: excellent

Author:  Jean Michel Maillet  on 2017-02-18  [id 101]

(in reply to Report 2 on 2017-01-13)

Thanks for the remark. Although we are not using it directly in our article, we added the reference [52] to the nice work of A. Doikou (2006) in the introduction.

---

## Round 2 · Referee Report · Anonymous (Referee 1) · 2017-1-13

Strengths

1 - the paper addresses quite complicated problem of investigation the boundary quantum integrable model at the root of unity
2 - developing of SoV method for the reflection algebra is an important problem, which was solved in the paper for the models associated to the Bazhanov-Strovanov L-operator

Weaknesses

1 - no weaknesses can be mentioned

Report

This paper is devoted to the investigation of the transfer matrix eigenvalue problem for the
class of cyclic 6-vertex representations of the reflection algebra in the framework of SoV
method and in the cases of one completely general and one triangular boundary matrices.
The paper follows the original Sklyanin's approach developed for boundary XXZ spin 1/2 quantum chain.
The paper is quite technical but is clearly written and present new results on the quantum
integrable models at the root of unity. Taking this into account, I would like to state that the paper
"Transfer matrix spectrum for cyclic representations of the 6-vertex reflection algebra I"
by J.M. Maillet, G. Niccoli and B. Pezelier can be published in the SciPost Physics Journal
if some minor changes listed below will be introduced.

Requested changes

In Section 3 authors are saying about the most general cyclic representations of the 6-vertex
reflection algebra and the most general scalar solution for the boundary matrix. It seems that there are lack of arguments why these are most general solutions of reference to the paper where such arguments were presented.

In the sentence which include display formulas (3.20), (3.21) and (3.22) there is a discrepancy
in the introduction of the coefficients $a_+(\lambda)$ and $d_+(\lambda)$ in (3.20) and there definition in (3.21) and (3.22). It seems that authors used different LaTeX fonts for the same objects.

What is a difference between a ${\cal B}_-(\lambda)$-eigenstates basis introduced
by (4.24), (4.25), (4.36), (4.37) and vectors introduced in (4.46)? If they are the same why different fonts were used, namely, boldface italic in the first case and simple mathbold in the second? If they are different, it is reasonable to comment what is a difference?

What does mean the notation $M_{\varkappa({h})\varkappa({h})}$ used in (4.50)
to characterise the matrix $M\equiv U^{(L)}U^{(R)}$? I was unable to find using of this notation in the rest of the paper.

There are primes in the formula (6.3) although it is said in remark after (5.14) that "In the following we will suppress the unnecessary prime in $\kappa_+$ and $\tau_+$." Is it correct?

  • validity: high
  • significance: high
  • originality: high
  • clarity: high
  • formatting: excellent
  • grammar: excellent

Author:  Jean Michel Maillet  on 2017-02-18  [id 102]

(in reply to Report 1 on 2017-01-13)

First we wish to thank the referee for his careful reading of the manuscript. More precisely, following the comments and questions raised :

Section 3 :

a) - The property we wanted to emphasize was indeed not stated in an enough precise manner; the K-matrix we use is the most general solution of reflection equation associated to the 6-vertex R-matrix; so the cyclic reflection algebra representations we consider are the most general one's associated to the Bazhanov-Stroganov Lax-operator. There exists indeed more general cyclic representations and the corresponding sentence has been modified to avoid any misunderstanding in section 3 and also in the end of the introduction. Thanks for this remark. We also added references for cyclic representations [57-61].

b) - The quantities $a_{+}$ and $d_{+}$ in (3.20) are defined in (3.3) and in (3.7) as elements of the K-matrix. The quantities $\mathsf{a}_{+}$ and $\mathsf{d}_{+}$ are completely different quantities defined in (3.21, 3.22), hence the different fonts used.

Section 4 :

a) - There is no difference between eigenstates in (4.24), (4.25), (4.36), (4.37) and in (4.46), both are indeed $\mathcal{B}_{-}(\lambda )$-eigenstates. Thanks for noticing this font problem. Corrected in the new version (all are now in italic style).

b) - The $p^{\mathsf{N}}\times p^{\mathsf{N}}$ matrix $M\equiv U^{(L)}U^{(R)}$ is the matrix of scalar products of left and right $\mathcal{B}_{-}$-eigenstates. To label its entries we use the $\varkappa$ map defined in (4.47). We prove that this matrix is diagonal and that (4.50) gives the values of its diagonal entries. We don't use this notation later on in the article, but we use the result stated in Proposition 4.1, for example in Proposition 4.2 to compute the scalar products of separate states.

Section 6 :

According to the remark after (5.14) the primes should indeed not be present in formula (6.3); thanks for the notice. Corrected in new version.

---

## Round 3 · Author Response

This is the new version of our article including changes according to the referees reports and our replies to these reports.

---

## Round 3 · List of Changes

See replies to referees reports where all changes are listed.

---

## Editorial Decision

published